# Semantic-Syntactic Discrepancy in Images (SSDI): Learning Meaning and Order of Features from Natural Images

**Chun Tao***                                              *tao88@purdue.edu*
*Department of Electrical and Computer Engineering*
*Purdue University*

**Timur Ibrayev***                                          *tibrayev@purdue.edu*
*Department of Electrical and Computer Engineering*
*Purdue University*

**Kaushik Roy**                                             *kaushik@purdue.edu*
*Department of Electrical and Computer Engineering*
*Purdue University*

**Reviewed on OpenReview:** *https://openreview.net/forum?id=8otbGorZK2*

## Abstract

Despite considerable progress in image classification tasks, classification models seem unaffected by the images that significantly deviate from those that appear natural to human eyes. Specifically, while human perception can easily identify abnormal appearances or compositions in images, classification models overlook any alterations in the arrangement of object parts as long as they are present in any order, even if unnatural. Hence, this work exposes the vulnerability of having semantic and syntactic discrepancy in images (SSDI) in the form of corruptions that remove or shuffle image patches or present images in the form of puzzles. To address this vulnerability, we propose the concept of "image grammar", comprising "image semantics" and "image syntax". Image semantics pertains to the interpretation of parts or patches within an image, whereas image syntax refers to the arrangement of these parts to form a coherent object. We present a semi-supervised two-stage method for learning the image grammar of visual elements and environments solely from natural images. While the first stage learns the semantic meaning of individual object parts, the second stage learns how their relative arrangement constitutes an entire object. The efficacy of the proposed approach is then demonstrated by achieving SSDI detection rates ranging from 70% to 90% on corruptions generated from CelebA and SUN-RGBD datasets. Code is publicly available at: https://github.com/ChunTao1999/SSDI/.

## 1 Introduction

The task of image classification has significantly evolved with the advancements in deep neural networks (DNNs), to the point of achieving performance comparable to humans. For such tasks, the state-of-the-art (SoTA) models are based on convolutional neural networks (CNNs) (Krizhevsky et al., 2012; Simonyan & Zisserman, 2015; He et al., 2016) and vision transformers (ViTs) (Dosovitskiy et al., 2021; Liu et al., 2021). The fundamental idea behind these models was to develop the ability to identify an object in an image by understanding (and recognizing) its features/attributes (semantics). For example, when an image is classified as a "dog", it is because it contains attributes of dogs, such as "fur", "dog tail", "dog paws". Indeed, such models learn the distribution of images belonging to different object classes by extracting

---

*These authors contributed equally and share first authorship.

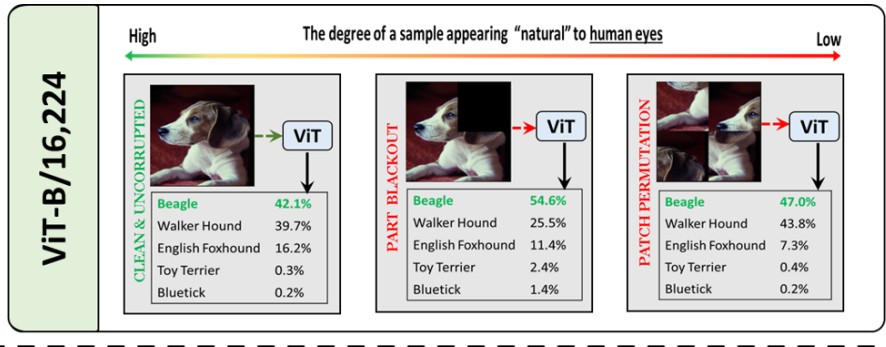

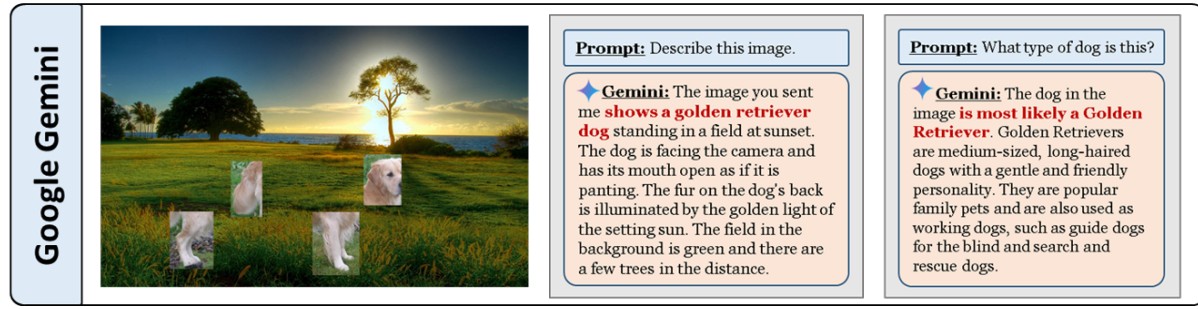

Figure 1: Examples illustrating susceptibility of various models (ViT-B/16,224 and Google Gemini Advanced) to images with unnatural appearance. The outputs of both models seem **to be completely unaffected by any of the abnormalities** present in the input images when performing the given tasks.

low- and high-level semantic information from images and encoding them in hidden features of stacked layers (Hua et al., 2018; Ortego et al., 2021). The learning processes are designed to mainly rely on image-level labels: the global information about the object type depicted in the entire image. Hence, to learn object classes, the classification models **have to implicitly learn** class features/attributes present in the images labeled as the corresponding class. In other words, since there is no explicit information provided for each feature like "fur", "paws", or "dog tail", the models must implicitly learn these attributes as characteristic traits of a dog by observing a large variety of "dog" images. However, this learning process has an underlying assumption that causes a vulnerability in the pipeline of classification models.

The vulnerability arises from assuming that ***input images always accurately depict the object in its natural appearance and composition***. As a result, classification models do not explicitly learn how the constituent features of an object class are arranged in natural images. For instance, they do not ensure that the "dog tail" is at the posterior end of the dog, or that the "paws" are in the lower part of the dog's body. In other words, the question being answered is "*What constituent class features are present in the image?*", and not "*Does this image contain a meaningful object of any observed class?*". This vulnerability appears as **the discrepancy** between a human's ability to **instinctively** recognize natural images **and** a classification model's tendency to be easily **fooled** into making high-confidence predictions for unnatural images. Figure 1 illustrates how a vision transformer model (ViT-B/16,224) (Wu et al., 2020) and a large multimodal model (Google Gemini Advanced) (Team et al., 2023) are susceptible to this type of vulnerability. It can be seen that the ViT model consistently predicts various dog breeds as the top-5 predictions, disregarding the appearance of images that increasingly diverge from the clean natural image. Similarly, the Gemini model ignores the unnaturalness of the dog displayed in the image when asked to describe the image or identify the type of the dog. We use the term **Semantic-Syntactic Discrepancy in Images (SSDI)** to refer to this vulnerability.

To address this vulnerability to SSDI corruptions in the visual appearance of input samples, we propose the concept of "***image grammar***". The concept is akin to grammar in language (Gunter et al., 1997; von Stechow, 2019), and comprises both semantic meaning ("***image semantics***") and a syntactical structure ("***image syntax***"). "*Image semantics*" pertains to the existence and semantic significance of individual

features defining an object, while "*image syntax*" concerns the spatial arrangement and correct placement of features to depict an object as it would naturally appear in the real world.

Consequently, we propose a semi-supervised two-stage ***deep learning framework*** that successively learns both "*image semantics*" and "*image syntax*" mentioned above. This framework combines a deep clustering method with a bi-directional LSTM. The deep clustering method treats the image as a set of semantic features, while the bi-directional LSTM ensures that the features are learned in relation to their spatial neighbors, enforcing the model's understanding of individual features and their natural arrangement and composition. This approach aims to train a classification model to not only predict object classes but also assess the degree of naturalness in the appearance of an image. Furthermore, the framework is specifically designed to learn the concept of "*image grammar*" in a strict setting that assumes SSDI corruptions are neither generated nor used during the training phase. Such a setting is motivated by the idea that, similar to human capabilities, the meaning and the order of features should be inherently learned together solely from natural images.

## 2 Semantic-Syntactic Discrepancy in Images (SSDI)

### 2.1 Problem Definition and SSDI Types

*Semantic-syntactic discrepancy (SSDI)* occurs in images that have *visually identifiable semantic features* but look unnatural to humans in their appearance and composition *due to incorrect arrangement or missing some of those features.* Figure 2 illustrates examples that expose the described vulnerability of classification models. As humans, we can readily recognize that the shown images aim to represent dogs due to the presence of recognizable dog features. We can also perceive that the images in the three rightmost columns ((c)–(e)) are more likely to be unnatural or altered in appearance (compared to images in columns (a) and (b)). We can make such a distinction even, for example, with a "puzzle" image with rearranged patches that we have not encountered before. In contrast, a classification model trained on large-scale dataset like ImageNet (Deng et al., 2009) consistently assigns output labels corresponding to various dog breeds for the images in columns (b)–(e) solely based on the presence of identifying semantic features (the semantics), disregarding any abnormalities in the overall appearance (the syntax).

In this work, we explore three types of corruptions causing SSDI. The first corruption is *patch shuffling*, where a subset of image patches is randomly swapped. The second corruption is *patch blackening*, where a subset of image patches is blackened out. The third corruption is *puzzle solving*, where a subset of image patches is swapped, similarly to the patch shuffling. However, unlike patch shuffling, in puzzle solving, all possible permutations are generated for the specified number of image patches allowed for corruption. Puzzle solving is then assessed based on the capability to determine the original image out of all its variations. These corruptions set the basic framework for image manipulations inducing semantic-syntactic discrepancy in images: rearranging object parts but keeping them visible enough to identify the original object. While they might not define the exhaustive list of all such transformations, they serve as a simple exemplar to illustrate the strong effects of feature presence on the decision-making of various models regardless of how natural composition of object parts changes within the image.

### 2.2 Pervasiveness of SSDI in the Existing Methods

As a part of this work, the initial set of experiments is to evaluate the performance of the various image processing methods against samples with semantic-syntactic discrepancies. Two types of methods were considered: relying only on image processing as well as on the combination of images and language (also known as vision-language models, VLMs). These experiments have three major goals. The first goal is to validate the presence of the vulnerability by illustrating the cases when patch shuffling and patch blackening have minimal impact on model performances despite significantly impactful changes to the image syntax. The second goal is to show when such manipulations become SSDI corruptions. In particular, through the combination of quantitative results and qualitative assessment of the corresponding samples, we will illustrate the range of corrupted image patches (in terms of the size and their count) that makes the corrupt image fall under the definition of SSDI given in Section 2.1. The third goal is to demonstrate that even

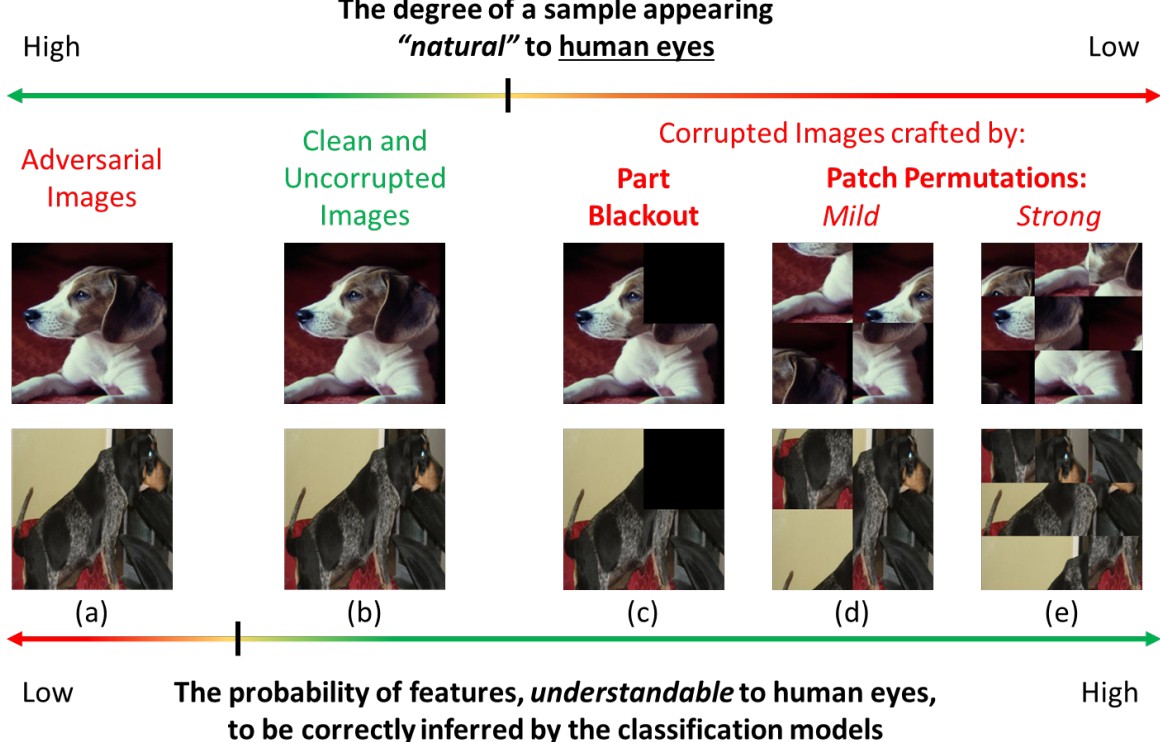

Figure 2: Examples explaining the vulnerability of classification models to semantic-syntactic discrepancies in images (SSDI). While all images appear to display different dogs, the visual appearance and composition of the images in columns (c)–(e) deviate from clean and uncorrupted images shown in column (b). This discrepancy, however, is not caught by the current classification models, which solely focus on the presence of features. This can be contrasted with adversarial examples (a), which target the decision-making of models while preserving the natural visual appearance. Note that adversarial examples are not considered in this study but are used to paint a complete picture.

multimodal models are vulnerable in their own ways to SSDI-type corruptions by not explicitly reflecting on any abnormalities unless explicitly requested by the user.

### 2.2.1 Purely Image-based Methods

Table 1 and Table 2 illustrate the classification accuracy of ViT-B-16 model on ImageNet2012 val samples under patch shuffling and patch blackening, respectively. Various patch sizes and numbers of patches were used to generate corrupted samples. With the original images being of size $224 \times 224$ pixels, it is possible to divide each image into patches of different sizes as represented by column headings. Consequently, there are different numbers of patches that are available for the corresponding SSDI type as represented by row headings. The selection of which patches are corrupted happens at random. Hence, in order to account for randomness, each cell represents the average accuracy of 5 experiments with different random seeding. For example, as shown in Table 2, the model accuracy drops from 81.07% base accuracy to an average of 69.16% when 2 out of 4 patches of size $112 \times 112$ were removed from every sample in ImageNet2012 val set. Appendix C contains tables illustrating performance results along with standard deviations for ViT-B-16 and ResNet-50 models, whose behavior matches that of the ViT model.

Based on the quantitative results of classification accuracy alone, the following observations can be made. First, for patch sizes equal to and smaller than $14 \times 14$ pixels (left halves of the tables), the model performance degrades drastically as a greater number of patches are corrupted. For patch blackening, the degradation

Table 1: Performance of ViT-B-16 model on ImageNet2012 with **"patch shuffling"** SSDI corruptions.

| | | Patch Size | | | | | | | | | |
|---|---|---|---|---|---|---|---|---|---|---|---|
| | | 12544 patches of $2 \times 2$ | 3136 patches of $4 \times 4$ | 1024 patches of $7 \times 7$ | 784 patches of $8 \times 8$ | 256 patches of $14 \times 14$ | 196 patches of $16 \times 16$ | 64 patches of $28 \times 28$ | 49 patches of $32 \times 32$ | 16 patches of $56 \times 56$ | 4 patches of $112 \times 112$ |
| | 0 | 81.07 | 81.07 | 81.07 | 81.07 | 81.07 | 81.07 | 81.07 | 81.07 | 81.07 | 81.07 |
| | 2 | 80.99 | 80.94 | 80.86 | 80.94 | 80.80 | 80.87 | 80.49 | 80.50 | 79.54 | **77.68*** |
| | 4 | 80.98 | 80.85 | 80.71 | 80.83 | 80.54 | 80.67 | 79.93 | 79.96 | 78.04 | 76.41 |
| | 16 | 80.63 | 80.36 | 79.84 | 80.19 | 78.85 | 79.75 | 75.52 | **75.88*** | 70.39 | |
| | 32 | 80.40 | 79.73 | 78.70 | 79.32 | 76.12 | **78.04*** | 66.25 | 67.78 | | |
| | 49 | 80.11 | 79.11 | 77.42 | 78.28 | 72.23 | 75.69 | 55.70 | 62.27 | | |
| | 64 | 79.81 | 78.71 | 76.40 | **77.31*** | 67.72 | 73.00 | 51.08 | | | |
| Number of Corrupted Patches | 128 | 78.91 | 76.64 | 70.97 | 72.27 | 35.77 | 52.59 | | | | |
| | 196 | **77.99*** | 74.64 | 62.94 | 64.28 | 8.16 | 36.98 | | | | |
| | 256 | 77.19 | 72.59 | 53.52 | 54.29 | 4.13 | | | | | |
| | 512 | 73.44 | 61.71 | 8.11 | 4.86 | | | | | | |
| | 784 | 69.27 | 45.39 | 0.59 | 0.79 | | | | | | |
| | 1024 | 65.63 | 28.38 | 0.41 | | | | | | | |
| | 2048 | 49.99 | 0.65 | | | | | | | | |
| | 3136 | 33.89 | 0.24 | | | | | | | | |
| | 4096 | 21.45 | | | | | | | | | |
| | 8192 | 0.92 | | | | | | | | | |
| | 12544 | 0.29 | | | | | | | | | |

* Corresponding examples are shown in Figure 3

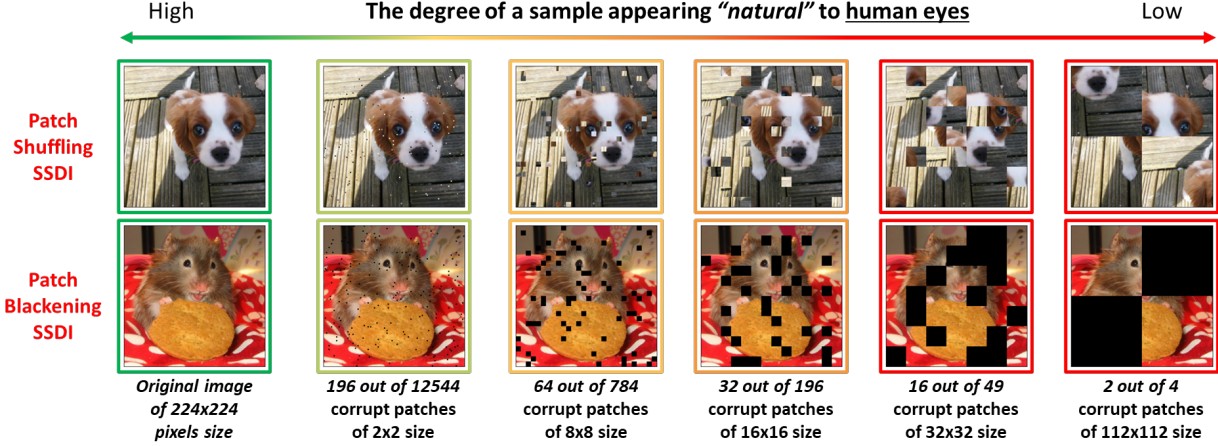

Figure 3: Examples of patch manipulations applied to randomly picked ImageNet2012 val set samples.

reaches random guessing (0.10%) since such image manipulations result in a trivial processing of black images. Second, for bigger patch sizes (above $16 \times 16$, right halves of the tables), the degradation is much more graceful. With the exception of random guessing in the case of patch blackening, modifications made to the samples are disregarded by the model in about half the cases. Finally, it can be observed that the accuracy remains relatively high compared to the base accuracy of 81.07% for a large portion of the table (green and yellow colored regions). While this behavior might have been favorable purely in terms of detecting object features, it also highlights that the model is largely "unbothered" by image manipulations.

Figure 3 demonstrates some random samples with different degrees of patch shuffling and patch blackening manipulations. Qualitative examination of these examples suggests explanations for the above quantitative observations. Image manipulations with small patch sizes ($2 \times 2$ and $8 \times 8$) are much less impactful in terms of both being perceptible to human eyes and affecting object parts as a whole. Instead, such changes would probably be perceived as noise both by humans and vision models, with its severity increasing with the number of corrupted image patches. Such cases would lie outside the definition of semantic-syntactic discrepancy, as the noise affects semantics more than syntax by "destroying/erasing" the object features, instead of changing their natural arrangement.

Table 2: Performance of ViT-B-16 model on ImageNet2012 with **"patch blackening"** SSDI corruptions.

| | | Patch Size | | | | | | | | | |
|---|---|---|---|---|---|---|---|---|---|---|---|
| | | 12544 patches of $2 \times 2$ | 3136 patches of $4 \times 4$ | 1024 patches of $7 \times 7$ | 784 patches of $8 \times 8$ | 256 patches of $14 \times 14$ | 196 patches of $16 \times 16$ | 64 patches of $28 \times 28$ | 49 patches of $32 \times 32$ | 16 patches of $56 \times 56$ | 4 patches of $112 \times 112$ |
| Number of Corrupted Patches | 0 | 81.07 | 81.07 | 81.07 | 81.07 | 81.07 | 81.07 | 81.07 | 81.07 | 81.07 | 81.07 |
| | 1 | 81.00 | 80.98 | 80.95 | 80.98 | 80.93 | 80.95 | 80.83 | 80.78 | 80.41 | 77.49 |
| | 2 | 80.99 | 80.96 | 80.86 | 80.97 | 80.82 | 80.84 | 80.62 | 80.49 | 79.55 | **69.16*** |
| | 4 | 80.96 | 80.85 | 80.77 | 80.88 | 80.67 | 80.65 | 80.26 | 79.75 | 77.26 | 0.10 |
| | 16 | 80.61 | 80.41 | 80.01 | 80.39 | 79.44 | 79.75 | 76.50 | **75.19*** | 0.10 | |
| | 32 | 80.43 | 79.94 | 79.16 | 79.65 | 77.41 | **78.60*** | 67.09 | 61.16 | | |
| | 49 | 80.24 | 79.51 | 78.20 | 78.89 | 74.69 | 77.27 | 43.36 | 0.10 | | |
| | 64 | 79.97 | 79.13 | 77.33 | **78.19*** | 71.54 | 75.86 | 0.10 | | | |
| | 128 | 79.30 | 77.60 | 73.14 | 74.50 | 46.11 | 61.75 | | | | |
| | 196 | **78.61*** | 75.87 | 67.53 | 69.02 | 9.08 | 0.10 | | | | |
| | 256 | 77.90 | 74.24 | 61.68 | 62.21 | 0.10 | | | | | |
| | 512 | 75.12 | 66.59 | 22.80 | 16.21 | | | | | | |
| | 784 | 72.40 | 56.67 | 0.78 | 0.10 | | | | | | |
| | 1024 | 70.23 | 46.28 | 0.10 | | | | | | | |
| | 2048 | 61.76 | 4.89 | | | | | | | | |
| | 3136 | 52.98 | 0.10 | | | | | | | | |
| | 4096 | 45.19 | | | | | | | | | |
| | 8192 | 9.88 | | | | | | | | | |
| | 12544 | 0.10 | | | | | | | | | |

* Corresponding examples are shown in Figure 3

On the other hand, the manipulations with the large patch sizes ($16 \times 16$ and above) do not necessarily target the fine-granularity of individual object features. Instead, they affect object features as a whole, such as obscuring "eyes" and "ears", or swapping "body parts". This explains why the accuracy results show no change even at the level of manipulations affecting up to half of the image. We acknowledge that the degree of naturalness is a subjective measure, similar to estimating saliency and attention in scenes, as it varies from person-to-person. However, we also highlight that the amount of changes induced by removing or swapping 2 *out of* 4 *patches of size* $112 \times 112$ is more conspicuous to human eyes than that at 64 *out of* 784 *patches of size* $8 \times 8$, both of which result in the same accuracy performance of the model. As a result, this reflects on the definition of SSDI corruptions. Specifically, for the given ImageNet2012 samples of size $224 \times 224$ pixels, patch shuffling and blackening at the patch sizes equal to and larger than $16 \times 16$ pixels cause images to appear unnatural while the underlying object parts remain visually identifiable.

### 2.2.2 Image and Language-based Methods

Table 3 illustrates the experimental results conducted on Meta's LLaMA large-scale vision-language model. Notably, LLaMA version 3.2 with 11 billion parameters available on HuggingFace hub was utilized. The experiment was conducted as follows. A subset of 3000 images was randomly selected from ImageNet2012 val set, such that there are 3 samples from each one of the thousand classes. Samples with original size $224 \times 224$ pixels were corrupted based on one of the ten configurations shown in the leftmost column. Then, these samples were processed by the multimodal model along with one of the three language prompts. These language prompts are:

**Prompt 1:** *What is the class of the object shown in the given image?*

**Prompt 2:** *Describe the given image.*

**Prompt 3:** *Does the given image appear natural or unnatural?*

Each of these prompts provides a different degree of guidance in terms of revealing the possibility of abnormalities being present in the given images. Prompt 1 only requests the class of an object, mimicking the behavior of the classifier. Prompt 2 requests a general description of an image. Prompt 3 implies that there are possible deviations present in the image. Please note that in all three cases the model is given complete freedom in its responses, such that no extra conditions are set (e.g., respond with a single sentence).

Table 3: Performance of Meta's LLaMA model on 3k samples of ImageNet2012 with various corruptions and with different language-based prompts. Each percentage value represents the percentage of LLaMA's responses that explicitly reflected that given samples looked "unnatural, corrupted, or tampered with". Please refer to the experiment details in Section 2.2.2.

| SSDI corruption (patch size, total # of patches, # of corrupted patches) | Prompt given to Meta's LLaMA model | | |
|---|---|---|---|
| | *What is the class of the object shown in the given image?* | *Describe the given image.* | *Does the given image appear natural or unnatural?* |
| **Patch Shuffling** | | | |
| $112 \times 112$, 4, 2 | 0.03% | 0.67% | 93.23% |
| $32 \times 32$, 49, 16 | 4.60% | 46.70% | 90.67% |
| $16 \times 16$, 196, 32 | 4.77% | 35.60% | 98.60% |
| $8 \times 8$, 784, 64 | 4.53% | 30.93% | 74.37% |
| $2 \times 2$, 12544, 196 | 0.27% | 8.03% | 97.20% |
| **Patch Blackening** | | | |
| $112 \times 112$, 4, 2 | 1.83% | 38.40% | 51.53% |
| $32 \times 32$, 49, 16 | 2.50% | 44.03% | 85.47% |
| $16 \times 16$, 196, 32 | 1.43% | 31.37% | 77.23% |
| $8 \times 8$, 784, 64 | 0.07% | 31.00% | 82.60% |
| $2 \times 2$, 12544, 196 | 0.00% | 33.03% | 89.03% |

After all pairs of corrupt images and prompts are processed by the model and its responses are collected, we utilized OpenAI's ChatGPT to evaluate the output responses. Specifically, ChatGPT evaluated Meta's LLaMA responses with the following evaluation prompt: *"Does the given response mention anything that indicates that the given image is unnatural, corrupted, or tampered with? Please answer with yes or no".* Based on such evaluation setup, Table 3 shows the percentage of samples for which the VLM **indicated that the given images were corrupted in some way**.

Based on these results of the vision-language model, the following observations can be made. When asked directly whether there are any abnormalities with the given images (with prompt 3), the detection rate of the VLM model is steadily high. This means that a prior needs to be set in order for the VLM to actively search for SSDI corruptions. On the other hand, interestingly, the detection rate is substantially lower in the other two cases. In case of prompt 1, it is indeed possible to argue that the model was not supposed to provide any extra observations. However, prompt 2 is the most generic, arguably serving as the go-to option for general applications. Under all generated corruptions, the VLM mentions their possibility in the images in fewer than half of the samples. As a result, it can be said that semantic-syntactic discrepancy remains a relevant issue for vision-language models, which manifests itself when no prior precautions or warnings are supplied through the correct prompting.

### 2.2.3 Puzzle Solving as a Proxy Task

Part of the problem with SSDI corruptions is that the current classification models do not have an explicit way of estimating the degree of naturalness of input images. Hence, we conducted an experiment in which puzzle solving SSDI corruption serves as a proxy for the corruption detection task. Particularly, as was described in Section 2.1, for the puzzle solving SSDI type we generate all the possible permutations of an image given the number of patches into which it is broken down. For ImageNet2012 val samples of original size $224 \times 224$ pixels, we only considered puzzle solving when images are divided into equally sized square patches of $112 \times 112$ pixels. As a result, for every image we generated 24 variations of the image resulting from all 4! permutations of image patches (including the original image).

Table 4: Performance of different models on ImageNet2012 samples with **"puzzle solving"** SSDI corruptions, illustrating the capabilities of each model at (1) predicting the class of an object shown in the given image and (2) determining the most natural arrangement of an image (i.e. the original image) out of all possibilities. Please refer to the experiment details in Section 2.2.3.

| | Prediction Accuracy | |
| --- | --- | --- |
| | Correct Class of an Object | Correct "Puzzle" Arrangement of an Image |
| ResNet-50 | 78.04% | 0.12% (59/50k) |
| ViT-B-16 | **78.98%** | **12.78% (6390/50k)** |
| Meta's LLaMA | N/A | 4.00% (120/3k) |

The models are then assessed by their ability to determine the original image with the natural arrangement of object parts out of all variations. For classification models (ResNet-50 and ViT-B-16), the prediction of the correct puzzle relied on the output (softmax) confidence scores. Specifically, all puzzles (including the original image) were processed by these models. For each image, the puzzle with the highest output confidence was chosen as the model's prediction for the image with the correct arrangement. Furthermore, the output class of the predicted puzzle also served as the model's prediction for the class of an object shown in the image. For the VLM model (Meta's LLaMA), all puzzles (including the original image) were given in a single request along with the following prompt: *"which one of these images appears to be the most natural? Give only the numeric value of the index of the image."*. To account for the cases when the model response includes anything more than the index, the responses were post-processed to verify and, if necessary, to clean the outputs to refer to one of the puzzle indices. The class labels were not requested as the language models would predict classes with open vocabulary.

Table 4 shows the performance of different models at correctly determining the image with the original natural arrangement of features/object parts. It can be seen that all models lack the capability to distinguish the original image from all of the possible patch shuffling variations. The highest accuracy at solving the puzzle was achieved with the vision transformer, which might be due to their inherent training procedures relying on processing images as a set of patches with positional encoding. The most interesting observation, however, is that despite the models failing to predict the image with the natural arrangement, there is almost no drop in the accuracy of object class predictions with respect to the base accuracies. Since detecting the presence of object features is enough to make the classification prediction, the significant difference between class and puzzle predictions demonstrates the corresponding gap between their comprehension of image semantics and image syntax. The main concerning implication is that the high classification accuracy means the models internally generate and rely on a similar set of features when processing the correct image and the image with any unnatural appearance.

## 2.3 Importance and Application

The ability to recognize and classify objects is a foundational task in computer vision. However, current classification models primarily focus on the presence of features rather than their spatial arrangement, making them vulnerable to SSDI corruptions. Addressing SSDI is important in safety-critical and real-world applications where models must not only identify objects but also evaluate their structural validity.

While our study uses controlled SSDI corruptions (e.g., patch shuffling, missing parts), similar issues arise naturally in many real-world scenarios, such as facial recognition security, medical imaging, and robotics. For example, face recognition systems often misidentify individuals when their features are partially occluded or misaligned (e.g., wearing masks, disguises, or under poor lighting). Consequently, if a model solely relies on feature presence, it may wrongly authenticate or misclassify identities. A similar example can be drawn in medical scenarios, where diagnostics rely on anatomical structures. Artifacts, occlusions, or unnatural feature arrangements can lead to false diagnoses if the model lacks awareness of natural structures. Such scenarios are especially prominent today, as generative models create increasingly realistic images, but sometimes fail to preserve natural image syntax.

As DNN models move beyond simple classification tasks toward visual reasoning and multimodal learning, addressing SSDI is crucial. Our experimental results show that even advanced vision-language models struggle to detect SSDI inconsistencies, failing to flag unnatural images unless explicitly prompted. This underscores the broader need to develop a learning approach that captures both semantics and syntactic structures in the visual domain.

SSDI vulnerability represents a distinct failure mode that is not captured by existing robustness paradigms, such as common corruptions, adversarial attacks, and out-of-distribution (OoD) shifts. Unlike common corruptions, SSDI does not degrade images randomly but instead disrupts the spatial integrity of object features. A shuffled or misarranged object may retain all its semantic components, yet fail to represent a meaningful whole. Unlike adversarial attacks, SSDI does not involve imperceptible pixel-level perturbations. Instead, SSDI creates visible, unnatural compositions that models assign high-confidence predictions to, exposing their reliance on feature presence over structural coherence. Unlike OoD shifts, SSDI does not involve a semantic shift to a new domain. Instead, SSDI occurs within the same category as the training data but in an incorrect arrangement that does not appear in natural images.

## 2.4 Learning Setting

The contrast between human perception and classification models in these examples underscores ***three crucial properties*** needed to address this vulnerability: (1) classification models need to learn to estimate the degree to which an image depicts a natural occurrence in the real world, (2) based on this degree, they need to be able to distinguish between natural and unnatural images, and (3) preferably, they should be capable of learning this based solely on natural images.

This work proposes a way to enable classification models with properties (1) and (2) in a learning setting motivated by property (3). In particular, the method describes how the architecture of a classification model and its training can be modified to allow it to distinguish between natural and unnatural images by only observing natural unaltered images during training. This setting is based on the combination of two reasons. First, humans can instinctively recognize unnaturalness of different kinds even when they never came across such examples. Both the meaning and the order of features are learned together without any need to see images with some parts removed or shuffled. Second, it is challenging to consider and train on all the possible ways samples with semantic-syntactic discrepancy can be crafted (including, but not limited to corruptions described in this work). For example, if an image is divided into $n \times n$ patches, assuming that only the original image depicts a meaningful object, the number of unnatural samples becomes $(n^2! - 1)$. Not only does this become intractable for values of $n > 3$, but also has no direct way of ensuring that all of these samples do not depict meaningful objects. Hence, it is more intuitive to teach "*What the natural arrangement of features should look like?*" rather than "*How natural and unnatural images differ?*".

## 3 Related Works

To the best of the authors' knowledge, the problem of SSDI vulnerability is not yet well-established. This makes it difficult to contextualize it as a standalone research direction. Nevertheless, the issue of SSDI can be considered in the context of two categories of works: from the perspective of learning the compositionality of visual representations and from the perspective of vulnerabilities in computer vision.

### 3.1 Compositionality of Image Representations

Learning image representations creates a basis for visual tasks, such as image classification (He et al., 2016) and object detection (He et al., 2017; Lin et al., 2017). Although various supervised (Krizhevsky et al., 2012) and unsupervised/self-supervised (He et al., 2022) methods have been proposed to learn the underlying distribution of image features, reliance purely on the presence of features has recently become more prominent (Thrush et al., 2022). In the work of Yuksekgonul et al. (2023), the authors discuss and analyze the effects of naive reliance on contrastive pre-training and accuracy metrics. They conclude that large vision and language models, which currently serve as the basis for a wide range of applications, disregard composition, i.e., the underlying structure, of both visual and language inputs. Their findings support the ideas and

highlight the relevance of SSDI vulnerability discussed in this paper. The work of Qin et al. (2022) proposed a solution for ViT-based models (Wu et al., 2020) through patch-based negative augmentation. From this perspective, our work serves as an alternative approach, which proposes a way of learning compositionality along with the semantics of features *without the need for generation and reliance on negative samples* (as described in the Learning Setting subsection of Section 2). Even though our experiments focus on CNN-based models, it is possible to extend the method to vision transformers by considering them as the feature extractor model.

## 3.2 Vulnerability of Computer Vision

The SSDI vulnerability, where differences in the visual appearance of an image are noticeable to humans, stands in contrast to adversarial examples (Szegedy et al., 2013; Carlini & Wagner, 2017; Madry et al., 2017; Wang & He, 2021; Gubri et al., 2022; Zhang et al., 2023), where the adversarial image and the clean image appear visually similar to humans. Adversarial examples have been extensively studied in the works of Madry et al. (2017); Shah et al. (2023); Mo et al. (2022); Andriushchenko & Flammarion (2020), which explore methods to prevent detrimental changes in the output predictions of classification models. However, these approaches do not apply to SSDI, as they do not detect abnormalities in appearance. Moving away from the broader concept of adversarial examples, Hendrycks & Dietterich (2018) introduced a benchmark, ImageNet-C, for common corruptions and perturbations. While ImageNet-C addresses changes in the natural appearance of images through noise or color manipulations, it does not address cases where corruption stems from the rearrangement of natural feature order without affecting classifier predictions.

## 4 Methodology

In this section, we outline our proposed classification framework, which detects SSDI using a semi-supervised, two-stage approach. **The first stage**, which we refer to as *part semantics*, focuses on learning "***image semantics***". The idea is to create a pixel-wise segmentation map of the input image that produces a (limited) set of clusters, each representing meaningful object parts (object semantics). **The second stage** then utilizes the part semantics to learn how their relative arrangement constitutes an entire object, i.e., the "***image syntax***". The idea is for the model to learn the expected arrangement and composition *based on* part semantics obtainable from natural images. **During inference**, the deviation from the expected arrangement and the composition learned from natural images serves as the metric to detect SSDI.

**Stage 1: Learning Part Semantics**  As mentioned, the first stage focuses on learning the meaningful attributes of an object - its part semantics. To learn this, the input pixels are segmented (grouped) into a set of classes that represent individual object parts instead of the objects themselves. Although fully unsupervised (no labels) or weakly supervised (only object class labels) approaches are more appealing, our observations showed that these methods did not produce semantic maps with the desired level of detail and accuracy. These approaches only allow the separation of the entire object from the background (e.g., the entire face), but not its parts (e.g., nose, eyes, mouth). This is because there is a greater variation in color between the object and the background than among the parts within the object itself. Hence, this stage was implemented in a semi-supervised approach by using only a fraction of the ground-truth semantic segmentation maps with the deep clustering (Caron et al., 2018; Cho et al., 2021).

The training procedure is shown in Figure 4. Consider a set of input images $x_i, i = 1, \ldots, N$. Let a subset of them have pixel-level annotations $m_{ip}, i = 1, \ldots, M, M \ll N$, where $p$ stands for the $p$-th pixel. We assume that we have $C$ semantic classes in the pre-processed ground-truth (GT) annotations and $\forall (i, p), m_{ip} \in [0, 1, \ldots, C-1]$. The feature extractor has an embedding function $f_\theta$, which produces pixel-level feature vectors $z_{ip} = f_\theta(x_i)[p], \ z_{ip} \in \mathbb{R}^d$.

First, we fine-tune the DNN feature extractor under semi-supervision. Pixel-level embedding features are fed to a linear classifier $g_\omega, \ \omega \in \mathbb{R}^{d \times C}$ so that the feature dimensions are projected onto the number of semantic classes $C$. (Note: once the feature extractor has been fine-tuned, the classifier $g$ can be discarded.) For all images $x_i$ that have GT masks, the resulting pixel-level prediction becomes $g_\omega(z_{ip})$. The fine-tuning is then realized by minimizing the cross-entropy loss between pixel-level prediction $g_\omega(z_{ip})$ and GT segmentation

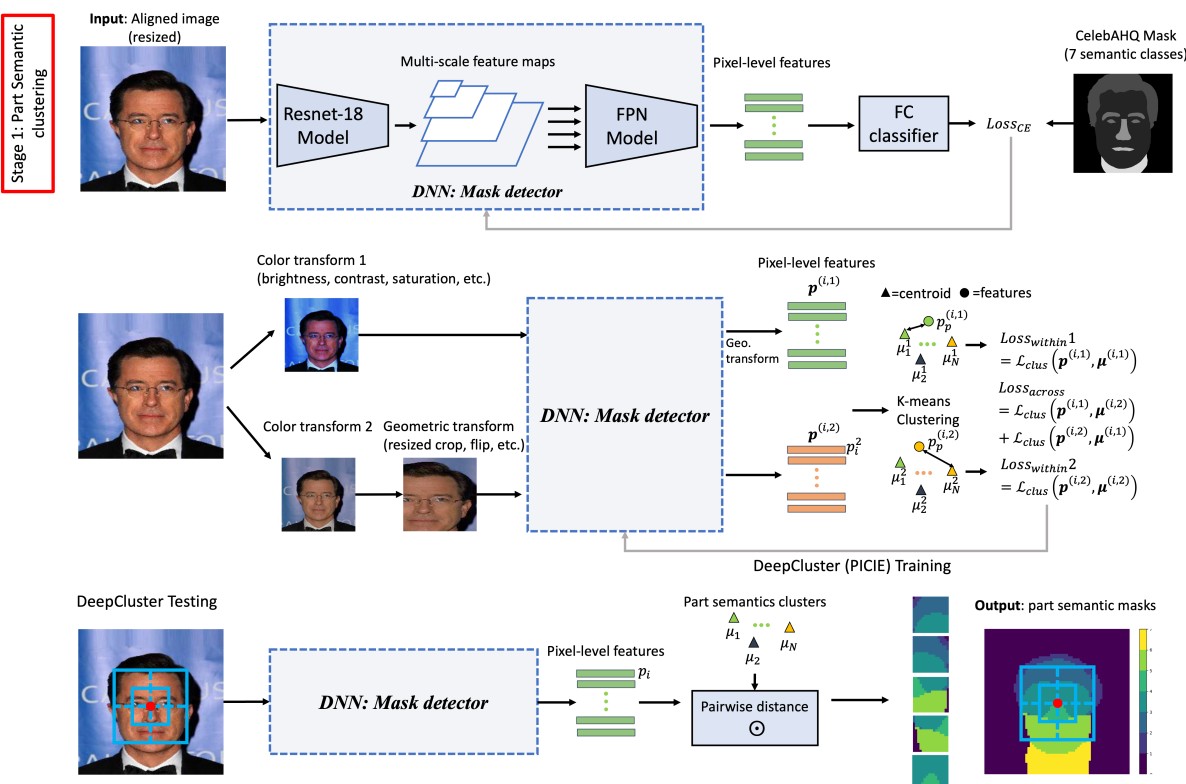

Figure 4: The first stage of training SSDI detection pipeline: learning part semantics of objects.

mask $m_{ip}$, as shown in Equation 1:

$$\min_{\theta,\omega} \mathcal{L}_{CE}(g_\omega(f_\theta(x_i)[p]), m_{ip}) \quad \text{where} \quad \mathcal{L}_{CE} = -\log \frac{\exp(g_\omega(z_{ip})[m_{ip}])}{\sum_{k=0}^{C-1} \exp(g_\omega(z_{ip})[k])} \tag{1}$$

As a second step, a deep clustering technique is used to expand upon semi-supervised knowledge. We used PiCIE, a deep clustering technique (Cho et al., 2021) in our approach. The core idea of PiCIE is to perform K-means clustering while allowing the model to account for inductive biases in the form of photometric invariance and geometric equivariance. In particular, photometric invariance accounts for the color variation (e.g., color jitter), whereas geometric equivariance accounts for the size and orientation variations (e.g., random crop) of semantic features in the given image. As shown in Figure 4, this is implemented by sampling each input into two transformation streams. In the first (upper) stream, an image $x_i$ is transformed according to photometric transformation $P^{(1)}$, processed by the model, and the resulting features transformed according to geometric transformation $G^{(1)}$. Contrary, in the second (bottom) stream, an image $x_i$ is transformed according to photometric transformation $P^{(2)}$, followed by geometric transformation $G^{(2)}$, and finally processed by the model. Geometric transforms $G^{(1)}$ and $G^{(2)}$ share the properties, such as the crop size, to ensure the resulting features are of the same dimensions. As a result, pixel-level features before K-means for two streams can be formulated as defined by Equations 2–3:

$$z_{ip}^{(1)} = G_i^{(1)}(f_\theta(P_i^{(1)}(x_i))) \tag{2}$$

$$z_{ip}^{(2)} = f_\theta(G_i^{(2)}(P_i^{(2)}(x_i))) \tag{3}$$

Given these pixel-level features, PiCIE optimizes the set of losses for K-means clustering as well as for inductive biases. Specifically, the K-means clustering is achieved by minimizing the loss defined by Equation 4:

$$\mathcal{L}_{Kmeans} = \sum_{i=1}^{N} \sum_{p} \sum_{k=1}^{K} \|z_{ipk} - \mu_k\|^2 \tag{4}$$

where $z_{ipk}$ indicates that the distance of each pixel-level feature is minimized with respect to their corresponding closest cluster centroid $\mu_k$.

The loss used to train inductive biases comprises two parts. The first part trains the model to extract pixel-level features that are consistently assigned to the same cluster centroid when a specific set of transformations is applied. In other words, if the same set of photometric ($P$) and geometric ($G$) transformations is used, this component of the loss ensures that the resulting features are as close as possible to the corresponding cluster centroid representing those features. Since this component solely considers the pixel-level features from each individual stream, it is denoted as $L_{within}$. However, this loss alone does not guarantee that the model will ignore variations when a different set of transformations is applied to the same image (i.e., it does not ensure invariance to photometric transformations or equivariance to geometric transformations). Therefore, the second part trains the model so that pixel-level features from two streams are assigned to the same cluster centroid even when different sets of transformations are applied. Because this component compares features between the two streams, it is denoted as $L_{cross}$. The total loss to minimize is $L_{total} = L_{within} + L_{cross}$, described by Equations 5, 6, and 7:

$$\mathcal{L}_{within} = \sum_{i=1}^{N} \sum_{p} \mathcal{L}_{DC}(z_{ip}^{(1)}, y_{ip}^{(1)}, \boldsymbol{\mu}) + \mathcal{L}_{DC}(z_{ip}^{(2)}, y_{ip}^{(2)}, \boldsymbol{\mu}) \tag{5}$$

$$\mathcal{L}_{cross} = \sum_{i=1}^{N} \sum_{p} \mathcal{L}_{DC}(z_{ip}^{(1)}, y_{ip}^{(2)}, \boldsymbol{\mu}) + \mathcal{L}_{DC}(z_{ip}^{(2)}, y_{ip}^{(1)}, \boldsymbol{\mu}) \tag{6}$$

$$\text{where} \quad \mathcal{L}_{DC}(z_{ip}, y, \boldsymbol{\mu}) = -\log \frac{\exp(-d(z_{ip}, \mu_y))}{\sum_{k=1}^{K} \exp(-d(z_{ip}, \mu_k))} \tag{7}$$

where $K$ is the total (pre-selected) number of clusters, $\boldsymbol{\mu}$ is the matrix of cluster centroids, $y$ is the centroid index in $\boldsymbol{\mu}$ that is closest to $z_{ip}$, and $d(\cdot, \cdot)$ is the cosine distance. As shown in Figure 4, the result of the first training stage is a DNN that can produce semantic masks for object parts, i.e., part semantics. The key is the desired granularity, such that the resulting part semantics are consistent for any given portion/patch of an image.

**Stage 2: Learning Image Syntax**   The second stage is the core of training the pipeline to detect SSDI. The idea is to utilize the relation between neighboring part semantics. For example, if we know that some portion of an image contains a human nose, the expectation is to have two eyes and a mouth depicted above and below it, respectively.

The first step in achieving this is to obtain different patches from within the input image. The works of FALcon (Ibrayev et al., 2023) and GFNet (Wang et al., 2020) explore processing input in a patch-wise sequential manner with only the class label given as the supervision. In this work, we consider a simple and generic approach to dividing images, i.e., the model on its own does not make any special decisions on how the patches are obtained. For the two datasets considered in this work, CelebA and SUN-RGBD, the patches are obtained as five crops around the center of the image (Figure 7) and as a zig-zag pattern from top to bottom, from left to right (Figure 9), respectively.

The second step is to traverse through the set of image patches, extract their corresponding part semantics, and learn the relation between the part semantics of neighboring patches. Suppose for image $x^{(i)}$, $i = 1, \ldots, N$, we extract a total of $G$ patches and obtain their corresponding part semantics $m_t^{(i)}$ for $t = 1, \ldots, G$ using the DNN trained in the first stage. The image syntax depends on the presence of the object parts with respect to each other rather than the exact part semantics $m_t^{(i)}$. Hence, each of them is converted into a part semantics vector $s_t^{(i)}$, which is the ratio of pixels belonging to each class in the part semantics.

Figure 5 shows how the image syntax is learned based on part semantics vectors using bidirectional long short-term memory (bi-LSTM). The bi-LSTM is used because both directions in the traversal of part semantics are valid relations to be learned. If it is parameterized by weights $\mathbf{W}$ and biases $\mathbf{b}$, then based on the inputs $m_t^{(i)}$ and $s_t^{(i)}$ as well as hidden states $h_t^{(i)}$ at the processing step $t$, the bi-LSTM makes two predictions: $p_t^{(i,\text{for})} = L_{\mathbf{W}, \mathbf{b}}(m_t^{(i)}, s_t^{(i)}, h_t^{(i,\text{for})})$ for the next semantics $s_{t+1}^{(i)}$ and $p_t^{(i,\text{back})} = L_{\mathbf{W}, \mathbf{b}}(m_t^{(i)}, s_t^{(i)}, h_t^{(i,\text{back})})$ for the previous semantics $s_{t-1}^{(i)}$. Across the set of training images $\mathbf{x}$, the learning goal of bi-LSTM is to capture the transition patterns between image part semantics. This is achieved by optimizing the mean squared error

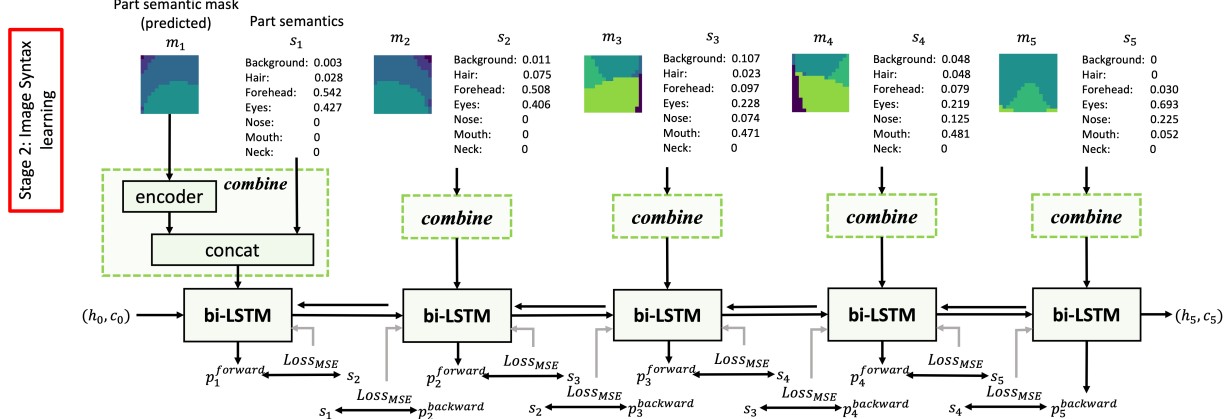

Figure 5: The second stage of training SSDI detection pipeline: learning image syntax.

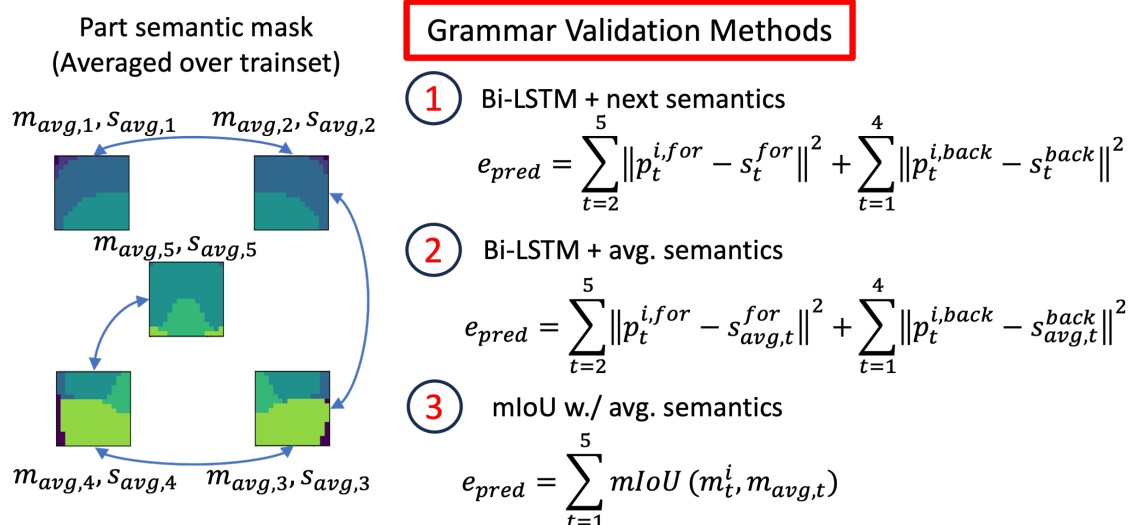

Figure 6: Three methods of quantifying the semantic-syntactic discrepancy in images, SSDI.

loss formulated by Equation 8:

$$\mathcal{L}_{MSE}(p_t^{(i)}, s_t^{(i)}) = \sum_{i=1}^{N} \left( \sum_{t=2}^{G} \|p_t^{(i,\text{for})} - s_t^{(i)}\|^2 + \sum_{t=1}^{G-1} \|p_t^{(i,\text{back})} - s_t^{(i)}\|^2 \right) \tag{8}$$

**Inference: Detecting SSDI** The detection of images with semantic-syntactic discrepancy (SSDI) is achieved by processing images in a similar patch-wise traversing manner and verifying whether the arrangement of object parts matches the expected arrangements learned from natural images. Figure 6 illustrates three methods of formally quantifying this process. The first method detects SSDI based on the relation of part semantics present in the image itself. Similar to the training process, for every image patch $t$, based on its part semantics, the bi-LSTM predicts the part semantics of its neighboring patches. The error is then computed as the average difference between the predicted part semantics and the part semantics estimated by the DNN.

The other two methods detect SSDI based on the memorization of the average part semantics of every pixel. Specifically, each pixel is assigned the most frequently predicted semantic class over the training set. For a test image, the error is then computed as the average difference between the predicted part semantics and

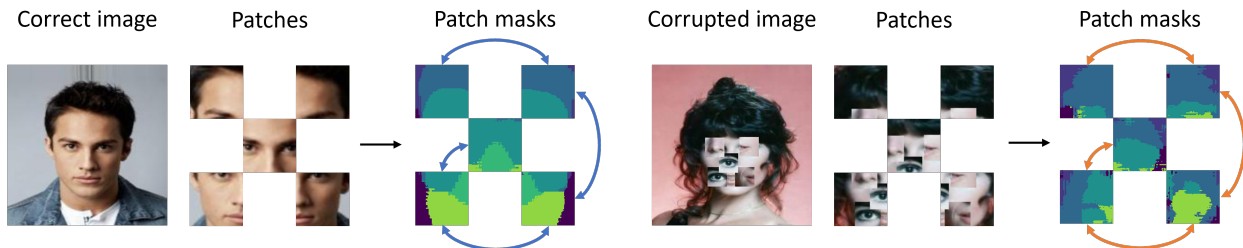

Figure 7: Image patches and the traversal sequences of processing image syntax of CelebA.

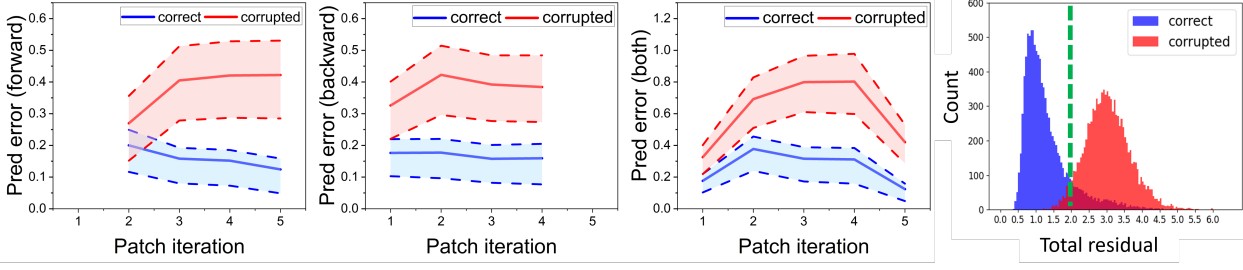

Figure 8: From left to right: the forward, the backward, and the total bi-LSTM prediction errors as well as the histogram of total errors over test samples of CelebA dataset.

the part semantics memorized based on the average semantics of training images. In the third validation method, the mean intersection over union (mIoU) is used instead of the difference, which is a common segmentation metric (Garcia-Garcia et al., 2017).

In all three cases, we obtain a quantitative measure of estimating the discrepancy in the semantic-syntactic relation in the form of computed errors. To use it as the detection during inference, we need to separate the error values for images with a natural and unnatural appearance. This is achieved by generating images with semantic-syntactic discrepancy and estimating thresholds for each method based on the validation set images. Specifically, we plot histograms of corrupted and original images from the validation set based on their total prediction errors. The threshold is then chosen as the error value that maximizes the separation between the two distributions. An example of this can be seen in Figure 8, where the green dashed vertical line marks the selected threshold.

## 5   Results

**Generation of SSDI and Performance Evaluation**   In this work, we consider three types of corruption causing semantic-syntactic discrepancy in images, as described in Section 2. For each considered dataset, SSDI corruptions are generated using half of the test set images chosen at random. All corruptions are considered separately, i.e., the paper does not consider a simultaneous mixture of different SSDI corruptions.

As described in Section 4, both natural and corrupted images with SSDI are processed by the framework in a patch-wise traversal manner, and the (residual) errors in predicting forward and backward part semantics by the bi-LSTM are computed. The threshold determined based on the validation set is used to distinguish between natural and corrupted images. We evaluate the performance based on two metrics: (1) the detection accuracy (i.e., binary classification accuracy) and (2) the detection rate (i.e., recall) described by Equation 9.

$$\text{Det. Acc.} = \frac{\text{TruePositive (TP)} + \text{TrueNegative (TN)}}{\text{FalseNegative (FN)} + \text{TP} + \text{FalsePositive (FP)} + \text{TN}} \quad \text{and} \quad \text{Det. Rate} = \frac{\text{TP}}{\text{FN} + \text{TP}} \quad (9)$$

**CelebA**   The first dataset is the aligned-and-cropped CelebA (Liu et al., 2015) dataset with 202,599 RGB images resized to $256 \times 256$. The DNN feature extractor consists of a Feature Pyramid Network

Table 5: The performance of the proposed approach on detecting SSDI corruptions on CelebA.

| Dataset | Part Semantics Model | Grammar Validation Method | Shuffle 2 20x20 | Shuffle 2 30x30 | Black 1 20x20 | Puzzles 4 20x20 | Puzzles 4 30x30 |
|---|---|---|---|---|---|---|---|
| *CelebA* | ResNet-18+FPN (ours) | Bi-LSTM + next semantics | 65.66 (68.67) | 76.22 (79.72) | 70.91 (67.04) | 86.74 | 92.51 |
| *CelebA* | ResNet-18+FPN (ours) | Bi-LSTM + avg. semantics | 61.53 (**71.72**) | 68.89 (**80.48**) | 65.84 (**72.38**) | 85.62 | 91.15 |
| *CelebA* | ResNet-18+FPN (ours) | mIoU w./ avg. semantics | 60.04 (56.16) | 69.59 (59.60) | 60.90 (61.87) | **99.25** | **99.58** |
| *CelebA* | SemanticGAN (Li et al., 2021) | Bi-LSTM + next semantics | 61.35 (**69.87**) | 70.20 (**73.68**) | 68.98 (**71.04**) | 82.37 | 88.09 |
| *CelebA* | SemanticGAN (Li et al., 2021) | Bi-LSTM + avg. semantics | 58.26 (67.91) | 63.57 (71.49) | 61.28 (65.78) | 81.23 | 86.79 |
| *CelebA* | SemanticGAN (Li et al., 2021) | mIoU w./ avg. semantics | 56.76 (58.34) | 62.75 (64.25) | 62.13 (61.87) | **98.60** | **99.12** |

(FPN) (Lin et al., 2017) with an ImageNet pre-trained ResNet-18 (He et al., 2016) as the backbone to extract $128 \times 64 \times 64$ downsampled pixel-level features. The FPN is fine-tuned with 30,000 CelebAHQ (Lee et al., 2020) face part segmentation masks (about 15% of CelebA). We perform mini-batch K-means clustering every 20 batches before a single centroid update. The number of clusters is set to 20. Part semantics masks were restricted to have 7 categories, i.e., every object is limited to being described by 7 object parts (including background) from selecting the top 7 clusters. For image syntax learning, a 1-layer bi-LSTM model is trained. Input and hidden vectors are 135-dimensional (128-d encoded mask concatenated with 7-d semantics), and the projected outputs are 7-d. Figure 7 shows the natural and unnatural images during the test, along with the 5 patches used in the traversal sequence and their corresponding predicted part semantics.

**SSDI Detection Performance on CelebA**   Figure 8 shows the forward, backward, and the sum of both residual errors of the bi-LSTM across iterations of processing image patches of both natural and corrupted images of the CelebA dataset. The solid lines are mean values, whereas the shaded region covers between the 25% and 75% percentiles. It can be seen that there is a separation between natural (blue) and corrupted SSDI (red) samples. The histogram shows the distribution of test samples based on the total residual error, along with the threshold (green dashed line) used to distinguish samples from natural and corrupted.

Table 5 illustrates the performance of the proposed framework on the CelebA dataset for various grammar validation methods. Different columns represent corruption types, with the number and the size of the affected image patches. For example, **Shuffle** 2 $30 \times 30$ means that 2 image patches of size $30 \times 30$ pixels are swapped to generate the patch shuffling corruption, whereas **Puzzles** 4 $20 \times 20$ means that the framework processes a batch generated from all possible permutations (including that of the original image) of 4 randomly selected patches of size $20 \times 20$ pixels. Both the detection accuracy and the detection rate results are reported, with the latter presented in parentheses.

When ResNet-18 and FPN are used as the models for extraction of part semantics, it can be seen that using the memorized average part semantics performs the best among the three grammar validation methods. This can be explained by the fact that in the CelebA dataset, the object is the human face with a very well-defined structure. Hence, it is more effective to learn not only the expected relation between the neighboring object parts (i.e., the image syntax) but also what those object parts are expected to be (i.e., the memorized average part semantics). The puzzle variation of SSDI corruption poses an interesting problem, since a large number of possible rearrangements of object parts observed at the same time significantly reduces the margin at which they can be distinguished based on the residual error. Interestingly, the framework achieves above 99% performance on puzzles, highlighting that the proposed approach indeed learned the correct composition of object parts.

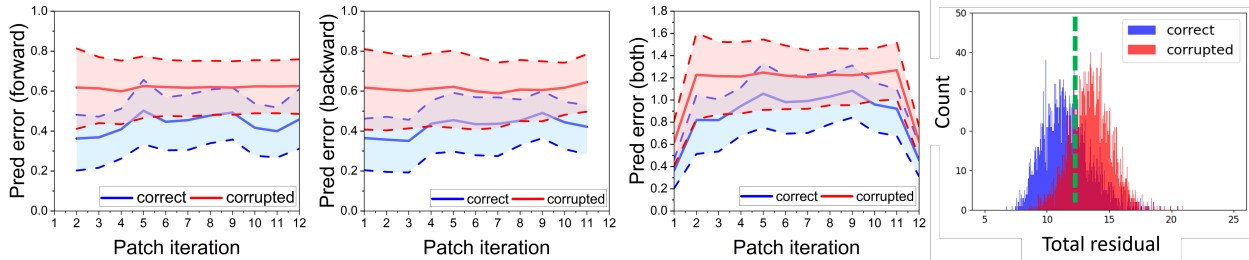

Figure 9: Image patches of size 160 and the sequences of processing image syntax of SUN-RGBD.

Figure 10: From left to right: the forward, the backward, and the total bi-LSTM prediction errors as well as the histogram of total errors over test samples of SUN-RGBD dataset.

**SUN-RGBD**  The second dataset is the SUN-RGBD (Song et al., 2015) dataset with 10,335 room layout RGB and depth images. The DNN feature extractor consists of a pre-trained Residual Encoder-Decoder (RedNet) (Jiang et al., 2018), which is used to retrieve part semantics masks from $640 \times 480$ sized images. The object parts were restricted to 13 based on the obtained semantic masks. For syntax learning, we use the same bi-LSTM model as in CelebA. As shown in Figure 9, we define zig-zag traversal rules on sequences of patches of sizes $160 \times 160$ and $80 \times 80$ pixels. The residual prediction errors and the histogram for test set images are shown in Figure 10.

Table 6 shows detection accuracy and detection rates on the SUN-RGBD dataset. Unlike the performance on the object-centric CelebA dataset, it is interesting to notice that the first grammar validation method is the most optimal for other corruptions on the scene-centric SUN-RGBD dataset. This can be explained by the more diverse nature of images in SUN-RGBD: different objects can be arranged in multiple different ways with respect to the entire scene and other objects. Hence, as scenes might change substantially, the method of using the stored average part semantics might be less useful than relying solely on the relations between object parts local to individual images.

**Effect of Segmentation Granularity**  On both datasets, we also consider more specialized alternatives for the segmentation masks, which are expected to produce more accurate and finer part semantics. The methods of SemanticGAN (Li et al., 2021) and Dformer (Yin et al., 2023) were considered as part semantics models for the CelebA and SUN-RGBD datasets, respectively. From test results presented in Table 5 and Table 6, it can be seen that finer semantic masks produced by more specialized segmentation methods do not always lead to higher SSDI detection rates, especially on an object-centric dataset like CelebA. This can probably be explained by the fact that it is easier to capture the natural arrangement of object parts if their part semantics is coarser. The relationship between semantic granularity and corruption detection deserves further investigation.

## 6 Limitations

While this work proposes a pioneering approach to tackle SSDI corruptions, there are a few limitations. First, the semi-supervised approach requires some fraction of semantic segmentation masks to obtain semantic separation of parts within the object, rather than objects from the background. A possible solution is to consider fully self-supervised segmentation methods, like DINO (Caron et al., 2021). Second, the current

Table 6: The performance of the proposed approach on detecting SSDI corruptions on SUN-RGBD.

| Dataset | Part Semantics Model | Grammar Validation Method | Shuffle 4 160x160 | Shuffle 16 80x80 | Black 4 160x160 | Puzzles 4 160x160 | Puzzles 16 80x80 |
|---|---|---|---|---|---|---|---|
| SUN-RGBD (13-cls.) | ResNet-50+Encoder-Decoder (ours) | Bi-LSTM + next semantics | 60.57 (**72.04**) | 76.57 (**73.37**) | 66.55 (**67.21**) | 72.89 | 91.17 |
| SUN-RGBD (13-cls.) | ResNet-50+Encoder-Decoder (ours) | Bi-LSTM + avg. semantics | 54.97 (65.36) | 61.52 (64.09) | 58.46 (59.65) | 62.47 | 75.48 |
| SUN-RGBD (13-cls.) | ResNet-50+Encoder-Decoder (ours) | mIoU w./ avg. semantics | 57.98 (68.04) | 58.98 (62.24) | 56.32 (57.19) | **97.09** | **98.20** |
| SUN-RGBD (13-cls.) | Dformer-S (Yin et al., 2023) | Bi-LSTM + next semantics | 62.16 (**73.47**) | 74.23 (**72.80**) | 68.86 (**71.23**) | 76.61 | 92.84 |
| SUN-RGBD (13-cls.) | Dformer-S (Yin et al., 2023) | Bi-LSTM + avg. semantics | 53.77 (67.23) | 62.45 (69.09) | 61.98 (62.45) | 61.97 | 74.78 |
| SUN-RGBD (13-cls.) | Dformer-S (Yin et al., 2023) | mIoU w./ avg. semantics | 53.24 (63.38) | 60.46 (66.21) | 59.79 (61.80) | **98.13** | **98.62** |

image syntax learning relies on the extraction and traversal of patches with a fixed pattern, which might limit its applicability to a variety of datasets. This shortcoming can be overcome by incorporating modules that learn "where to look" along with the proposed syntax learning approach, which do not have to rely on a fixed number and order of patches. Active vision methods, such as Saccader (Elsayed et al., 2019), GFNet (Wang et al., 2020), or FALcon (Ibrayev et al., 2023), provide appropriate frameworks that enable computer vision pipelines with the capability to process inputs through a series of glimpses. Since such active vision approaches rely on learning inherent clues between the sequence of glimpses, they might find interesting synergy with syntax learning, which naturally assumes the presence of such clues.

# 7 Conclusion

Motivated to bridge the gap between human and machine perception related to unnatural images, we introduce a novel deep learning framework for addressing semantic-syntactic discrepancy in images (SSDI). Aligned with the concept of language grammar, the framework learns both the meaning and the natural arrangement of object parts using deep clustering and a bi-directional LSTM. The effectiveness of the approach is shown through its capability of solving puzzles and achieving detection rates of 70–90% on CelebA and 60–80% on SUN-RGBD. The pioneering aspect of the problem suggests a broader impact in areas requiring reliable analysis of image content.

**Broader Impact Statement**

Machine learning systems and, particularly, computer vision techniques are having an increased influence on everyday tasks that benefit from automation. These tasks range from those with minimal impact on human life, such as facial recognition systems on mobile phones, to highly significant ones, such as obstacle detection in autonomous vehicles. A huge amount of trust is being placed in the learning frameworks that are used for performing these tasks. These frameworks are often trained on large datasets curated by retrieving images using machine learning recognition techniques. If the images gathered are unnatural (do not follow the syntax of real-world images), the learning of the downstream learning frameworks is also tainted. Hence, it is important for these learning frameworks to (a) be trained on data that accurately represents real-world scenarios and (b) learn the syntax and semantics of the data accurately. This work addresses both these concerns by highlighting (a) the SSDI vulnerability in datasets and the importance of learning "image grammar" consisting of "image syntax" and "image semantics" as well as (b) proposing a two-stage weakly supervised framework that attempts to minimize the possibility of the malicious use of these corruptions by learning the mentioned "image grammar". This paves the pathway for future researchers to implement computer vision techniques that operate holistically - learn both the syntax and semantics.

However, as in many examples in the history of digital systems, the exposure of vulnerabilities may lead to a path for malicious entities to exploit those vulnerabilities in existing systems. For example, when the vulnerability of computer vision techniques to additive imperceptible noise was exposed, many adversarial techniques were developed to trick existing machine learning frameworks. Consequently, there is a potential for developing attacks or malicious techniques that exploit the SSDI vulnerability in images for nefarious purposes. Moreover, the situation is worsened by the fact that such vulnerabilities are currently not detectable purely by machine learning, but would require a human expert in the loop for security. This further underscores the need for systems to wholly learn the "image syntax" along with the "image grammar".

**Acknowledgments**

This work was supported in part by, the Center for the Co-Design of Cognitive Systems (CoCoSys), a DARPA-sponsored JUMP 2.0 center, the Semiconductor Research Corporation (SRC), and the National Science Foundation.

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

# A  Experimental details

The entire framework was implemented using the PyTorch framework (Paszke et al., 2019). The models were trained and evaluated using 4 NVIDIA GeForce GTX 2080 Ti GPU cards.

## A.1  Datasets

CelebA (Liu et al., 2015) contains 202,599 face images. We used the aligned-and-cropped version where faces are localized. Images are divided into a train set of size 162,770, a validation set of size 19,867, and a test set of size 19,962. First, the FPN network is finetuned on 30,000 CelebAHQ (Lee et al., 2020) images selected from CelebA. CelebAHQ images and pixel-wise labels are center-cropped (size=160) and resized to $256 \times 256$. After fine-tuning, PiCIE deep clustering technique (Cho et al., 2021) is trained on train set images resized to $256 \times 256$ and validated on the validation set. $64 \times 64$ patches are cropped from segmentation and are upsampled using bilinear interpolation to $256 \times 256$. The 7 semantic classes and their names are illustrated on the left part of Figure 11. During the test, the corruptions are generated around 5 facial part landmark locations (left eye, right eye, nose, left mouth, right mouth) using a half of the randomly selected test images.

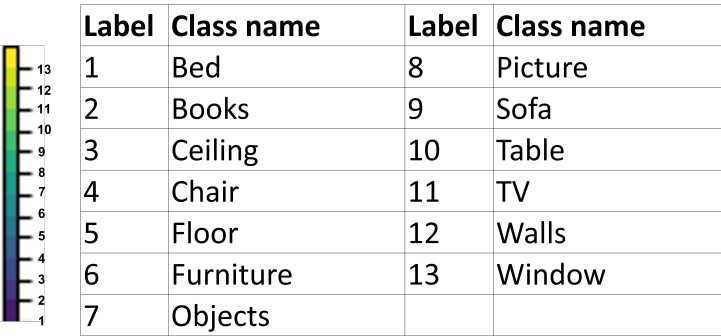

| Label | Class name |
|-------|------------|
| 0 | Background |
| 1 | Hair |
| 2 | Forehead |
| 3 | Eyes |
| 4 | Nose |
| 5 | Mouth |
| 6 | Neck |

| Label | Class name | Label | Class name |
|-------|------------|-------|------------|
| 1 | Bed | 8 | Picture |
| 2 | Books | 9 | Sofa |
| 3 | Ceiling | 10 | Table |
| 4 | Chair | 11 | TV |
| 5 | Floor | 12 | Walls |
| 6 | Furniture | 13 | Window |
| 7 | Objects | | |

Figure 11: Segmentation Classes. Left: CelebA; Right: SUN-RGBD

SUN-RGBD (Song et al., 2015) contains 10335 RGB images and corresponding depth images, split into a train set of size 4785, a validation set of size 1000, and a test set of size 5050. RGB and depth images are resized to $640 \times 480$ and normalized. The pre-trained RedNet (Jiang et al., 2018) also produces $640 \times 480$ part semantic masks. From each segmentation, $160 \times 160$ and $80 \times 80$ patch semantic masks are cropped. The segmentation contains 37 semantic classes, but we merge them into 13 classes with the same mapping used in the work of Handa et al. (2016). The 13 semantic classes are shown on the right part of Figure 11. During the test, we add corruptions to image patches using a half of the randomly selected test images.

## A.2  Architectures

**On CelebA**  Figure 12 illustrates the feature pyramid network (FPN) (Lin et al., 2017) that was used for part semantics clustering and segmentation of CelebA (Liu et al., 2015) images. Specifically, Panoptic FPN proposed in the work of Kirillov et al. (2019) was used in the combination with ResNet-18 (He et al., 2016) as the backbone feature extractor. While the backbone encodes multi-scale feature representation in a top-down fashion, FPN model utilizes $1 \times 1$ convolutional (Conv) layers in the bottom-up manner to project pixel-wise features to 128-dimensional space. The projected features of different spatial sizes are upsampled using bilinear interpolation to 1/4 height and width of the original image and then element-wise summed. The pixel-wise representations thus have shape $128 \times h \times w$ ($128 \times 64 \times 64$ for input images of shape $3 \times 256 \times 256$). The extracted pixel-wise features are used for the deep clustering (Cho et al., 2021). We record the resulting clustering centroids as a final $1 \times 1$ Conv layer to project these features to the dimension of the number of semantic classes to represent part semantics.

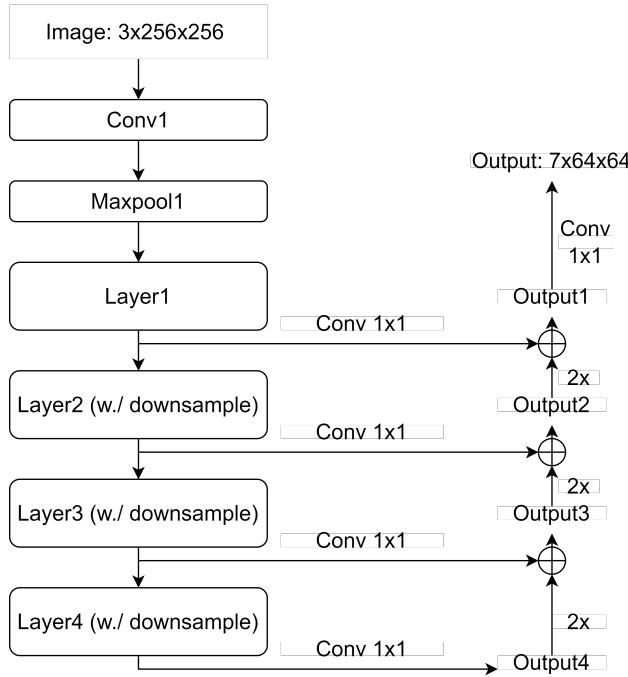

Figure 12: DNN architectures: Feature Pyramid Network (FPN) with ResNet-18 as the backbone

Figure 13 shows the structure of the bi-directional Long Short-Term Memory (bi-LSTM). We use it for image syntax learning. For each patch in the traversal sequence of length 5, the predicted mask is first flattened before being fed to a fully-connected (FC) encoder layer. The encoder projects the semantic mask to a 128-d vector, which is concatenated with the 7-d semantics vector. The resulting 135-d vector contains both spatial and semantic information about each patch's part semantics, and it is used as the input to 2 LSTM layers in forward and backward directions. After projection layers, the outputs include a 7-d forward prediction of the next part semantics semantics in the sequence and a 7-d backward prediction of the previous part semantics. For the first and last patches, only forward and backward predictions are made, respectively.

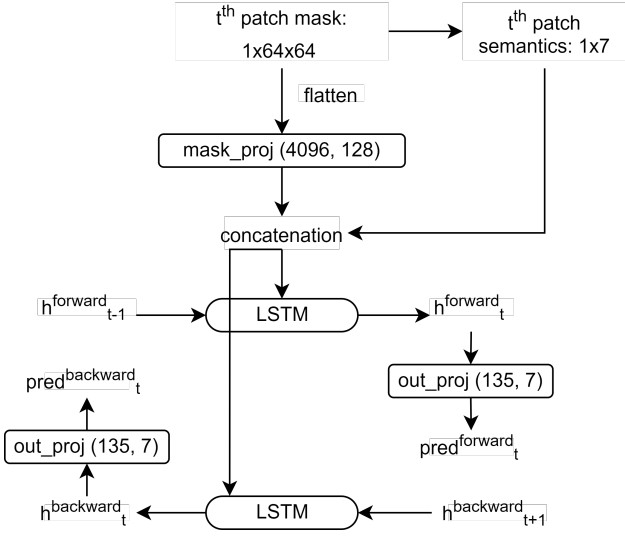

Figure 13: DNN architectures: bi-directional Long Short-Term Memory (bi-LSTM)

**On SUN-RGBD**   For segmentation of SUN-RGBD (Song et al., 2015) images, we use a Residual Encoder-Decoder Network (RedNet) (Jiang et al., 2018) with ResNet-50 as the backbone. It is a state-of-the-art room scene parsing architecture that achieved 47.8% mIoU accuracy on 37-class SUN-RGBD. Similar to FPN, the RedNet extracts multi-scale feature maps from the input image with its encoder residual layers. However, each projected feature map is skip-connected with the output of upsampling residual units in a decoder (Jiang et al., 2018) before being summed with the next level in the feature pyramid. An additional depth branch is fused with the RGB branch. We performed 13-class (merged from 37-class) semantic segmentation with a trained RedNet. The RedNet produces the same-resolution ($640 \times 480$) segmentation from the original image. Patch semantic masks of sizes $160 \times 160$ and $80 \times 80$ are cropped and used for preparing patch semantic vectors. At each stage of patch sequences, part semantics masks are flattened and projected to 128-d vectors before concatenating with a 13-d part semantics vector. The resulting 141-d vectors are passed to LSTM blocks for forward and backward predictions (both 13-d).

### A.3   Hyperparameters

**On CelebA**   During fine-tuning of the FPN network pre-trained on ImageNet (Deng et al., 2009), CelebAHQ (Lee et al., 2020) masks are pre-processed to contain only 7 coarse semantic classes. *Adam* optimizer is used with $start\_lr = 1e^{-4}$. We trained for $num\_epochs = 20$ with a constant learning rate and $batch\_size\_train = 128$. The fine-tuned model is used for deep clustering PiCIE (Cho et al., 2021) training with $batch\_size\_train = 256$. We adopted over-clustering and set the number of clusters, $K\_train = 20$, as that results in better clusters compared to $KM\_num = 10$ and $KM\_num = 30$. Mini-batch K-means clustering is performed. $KM\_init = 20$ is the number of batches we collect before the first K-means clustering. $KM\_num = 20$ is the interval of batches between consecutive clustering. $KM\_iter = 100$ is the number of K-means clustering iterations before convergence. After each clustering, the total PiCIE loss is back-propagated to optimize FPN model parameters. We use *Adam* optimizer with $start\_lr = 1e^{-4}$. Despite the complexity of the total PiCIE training loss, the training converges fast in $num\_epochs = 10$ with a constant learning rate. To get semantic segmentation masks, we initiate a $1 \times 1$ Conv layer, the weights of which are loaded as the trained centroid matrix, and append it to the trained FPN network. We post-process pixel-wise class assignments to merge semantically-similar clusters and obtain 7-class face part semantics. Further, we tune the FPN network on $64 \times 64$ patches, using crops on the saved face part semantics as the supervision. The part semantics mask detector is trained with $batch\_size\_train = 128$, $start\_lr = 1e^{-4}$ for a total of $num\_epochs = 20$. We chose the $num\_epochs = 20$ model over the $num\_epochs = 40$ model because the smooth boundaries in the semantic masks help with image syntax learning.

After we obtain 7-class part semantics masks for each patch in the traversal sequence, we feed the concatenation of encoded mask (128-d) and semantics vector (7-d) into a bi-LSTM model. The parameters for bi-LSTM are: $input\_size = 135$, $hidden\_size = 135$, $num\_layers = 1$, $bidirectional = True$, $proj\_size = 7$. We use *Adam* optimizer with $start\_lr = 1e^{-4}$ and $num\_epochs = 40$, with the learning rate changed to $1e^{-5}$ after 20 epochs.

**On SUN-RGBD**   On SUN-RGBD, the pre-trained RedNet network (Jiang et al., 2018) produces 37 segmentation classes. We merge these classes into 13-class as shown in Figure 11. When the patches are shuffled, the pre-trained network is able to produce precise part semantics, so that a part mask detector is not needed as in CelebA. We start with image syntax training with two configurations, $patch\_size = 160$ and $patch\_size = 80$. In both configurations, the part semantics masks are encoded as 128-d vectors, while the part semantics vector is 13-d. Thus, the bi-LSTM model has the configuration with $input\_size = 141$, $hidden\_size = 141$, $num\_layers = 1$, $bidirectional = True$, $proj\_size = 13$. We use *Adam* optimizer with $start\_lr = 1e^{-4}$ for a total of $num\_epochs = 40$. A multi-step learning rate scheduler is used with $gamma = 0.8$ and $milestones = [5, 10, 15, 20, 25, 30, 35]$.

## B    Generation of SSDI corruptions

One of the main contributions of this work is exposing the vulnerability of classification models to semantic-syntactic discrepancy in images (SSDI). We define three types of corruptions causing semantic-syntactic discrepancy in images and provide the algorithmic methodology of generating these SSDI corruptions. Furthermore, the supplementary material contains the code to generate the corresponding SSDI corruptions.

Algorithm 1 illustrates the algorithm used to generate the first type of SSDI corruption called a *patch shuffling*. This type of corruption is generated by swapping the positions of the subset of image patches. The performance of the SSDI detection approach is evaluated based on its capability to detect whether the given input is a natural image or not. Algorithm 2 illustrates the algorithm used to generate the second type of SSDI corruption called a *patch blackening*. This type of corruption is generated by blackening out a subset of image patches. Similarly to the *patch shuffling*, the performance is evaluated based on the detection capability of natural and corrupt images. Algorithm 3 illustrates the algorithm used to generate the third type of SSDI corruption called a *puzzle solving*. Similarly to the *patch shuffling*, this type of corruption permutes a subset of image patches. However, unlike patch shuffling, in puzzle solving, we generate all possible permutations for the specified number of image patches allowed for corruption. Then, all possible patch permutations and the original natural image are fed to the framework, which has to predict the image with the natural arrangement of image patches/object parts. For each considered dataset, SSDI corruptions are generated using half of the test set images chosen at random. All corruptions are considered separately, i.e. the paper does not consider a simultaneous mixture of different SSDI corruptions. Figure 14 illustrates examples of all SSDI corruptions considered in this work on CelebA and SUN-RGBD test images.

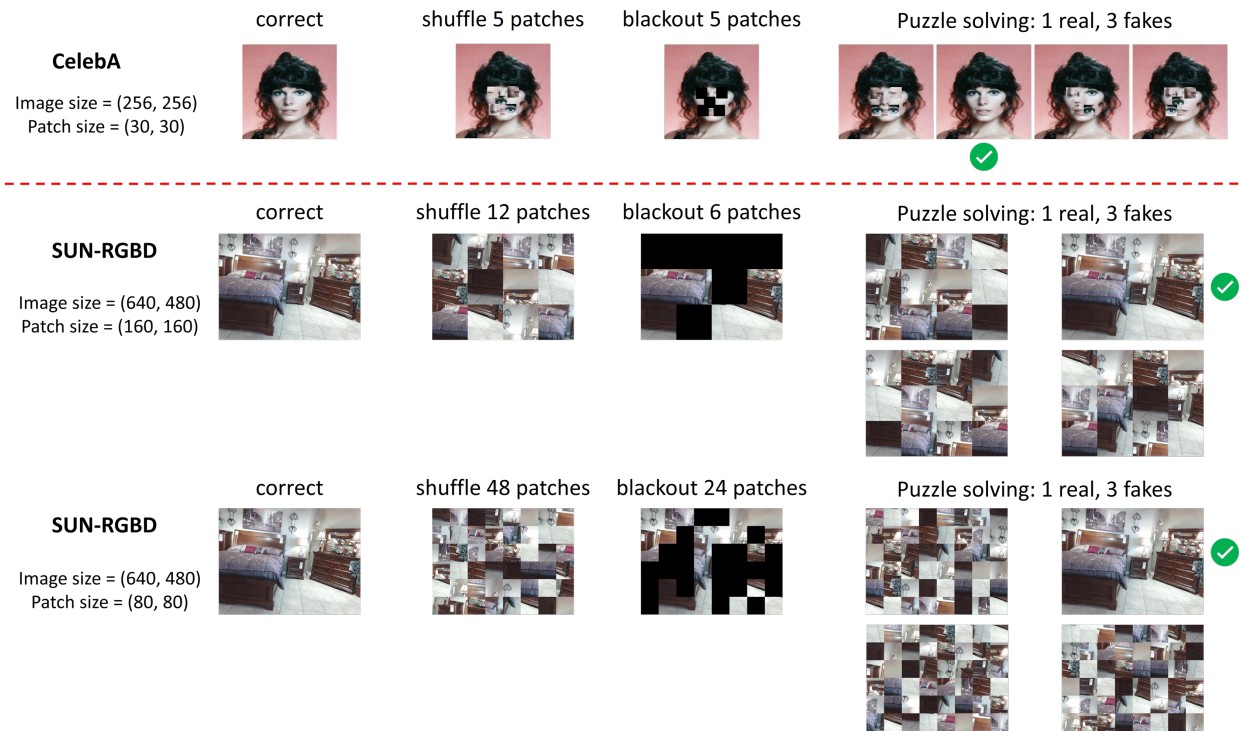

Figure 14: Demo of all types of SSDI corruptions.

---

**Algorithm 1** Patch shuffling

---

1: def shuffleSSDI(*tensors*, *num_patch*, *ps*):
2: $result \leftarrow []$
3: **for** $it, X$ in enumerate(*tensors*) **do**
4:     $patches \leftarrow$ nn.F.unfold($X$, *ps*, *ps*, 0)
5:     $p \leftarrow patches$.shape[-1]
6:     **if** $it == 0$ **then**
7:         $indices \leftarrow$ sample(range($p$), *num_patch*)
8:         $orig \leftarrow$ tensor(range(p))
9:         $perm \leftarrow$ tensor(range(p))
10:        **for** $j$ in range(num_patch) **do**
11:            $perm[indices[j]] = orig[indices[(j+1)\%num\_patch]]$
12:        **end for**
13:    **end if**
14:    $patches \leftarrow$ concat($patch[:,$ perm]for $patch$ in $patches$)
15:    $X \leftarrow$ nn.F.fold(patches, $X$.shape[-2:],*ps*, *ps*, 0)
16:    $result$.append($X$)
17: **end for**
18: **return** $result$

---

**Algorithm 2** Patch blackout

---

1: def blackoutSSDI(*tensors*, *num_patch*, *ps*):
2: $result \leftarrow []$
3: **for** $it, X$ in enumerate(*tensors*) **do**
4:     $patches \leftarrow$ nn.F.unfold($X$, *ps*, *ps*, 0)
5:     $p \leftarrow patches$.shape[-1]
6:     **if** $it == 0$ **then**
7:         $indices \leftarrow$ sample(range($p$), *num_patch*)
8:     **end if**
9:     **for** $idx, X$ in enumerate(*patches*) **do**
10:        $patches[idx][:, indices] \leftarrow 0$
11:    **end for**
12:    $X \leftarrow$ nn.F.fold(patches, $X$.shape[-2:],*ps*, *ps*, 0)
13:    $result$.append($X$)
14: **end for**
15: **return** $result$

---

---
**Algorithm 3** Create puzzles

---
1: def puzzleSSDI(*tensors*, *num_perm*, *ps*):
2:   *result* ← []
3:   **for** *it*, *X* in enumerate(*tensors*) **do**
4:     *patches* ← nn.F.unfold(*X*, *ps*, *ps*, 0)
5:     *p* ← *patches*.shape[-1]
6:     **if** *it* == 0 **then**
7:       *perms* ← []
8:       **for** *perm_id* in range(*num_perm*) **do**
9:         *perms*.append(randperm(*p*))
10:       **end for**
11:     **end if**
12:     *res* ← *X*.clone()
13:     **for** *perm* in *perms* **do**
14:       *new_patches* ← concat(*patch*[:, *perm*]for *patch* in *patches*)
15:       *new_X* ← nn.F.fold(patches, *X*.shape[-2:],*ps*, *ps*, 0)
16:       *res* ← concat(*res*, *new_X*)
17:     **end for**
18:     *result*.append(*res*)
19: **end for**
20: **return** *result*

---

# C   Performance results of classification models against patch corruptions

Figures 15, 16, 17, and 18 illustrate classification accuracy results achieved with different classification models against patch shuffling and patch blackening corruptions generated from ImageNet2012 val set samples of original size $224 \times 224$ pixels.

**Performance of ViT-B-16 model on ImageNet2012 with "patch shuffling" SSDI corruptions**
*(clean accuracy: 81.07%, original image size: 224x224)*

| Num Corrupted Patches | 4 patches of 112x112 | 16 patches of 56x56 | 49 patches of 32x32 | 64 patches of 28x28 | 196 patches of 16x16 | 256 patches of 14x14 | 784 patches of 8x8 | 1024 patches of 7x7 | 3136 patches of 4x4 | 12544 patches of 2x2 |
|---|---|---|---|---|---|---|---|---|---|---|
| 0 | 81.07 | 81.07 | 81.07 | 81.07 | 81.07 | 81.07 | 81.07 | 81.07 | 81.07 | 81.07 |
| 2 | 77.68±0.06 | 79.54±0.08 | 80.50±0.04 | 80.49±0.06 | 80.87±0.03 | 80.80±0.06 | 80.94±0.05 | 80.86±0.03 | 80.94±0.02 | 80.99±0.02 |
| 4 | 76.41±0.09 | 78.04±0.05 | 79.96±0.04 | 79.93±0.06 | 80.67±0.03 | 80.54±0.04 | 80.83±0.03 | 80.71±0.05 | 80.85±0.03 | 80.98±0.03 |
| 16 | | 70.39±0.14 | 75.88±0.08 | 75.52±0.08 | 79.75±0.08 | 78.85±0.06 | 80.19±0.04 | 79.84±0.06 | 80.36±0.06 | 80.63±0.05 |
| 32 | | | 67.78±0.11 | 66.25±0.05 | 78.04±0.13 | 76.12±0.09 | 79.32±0.08 | 78.70±0.08 | 79.73±0.08 | 80.40±0.02 |
| 49 | | | 62.27±0.05 | 55.70±0.16 | 75.69±0.06 | 72.23±0.09 | 78.28±0.10 | 77.42±0.07 | 79.11±0.05 | 80.11±0.02 |
| 64 | | | | 51.08±0.07 | 73.00±0.13 | 67.72±0.11 | 77.31±0.09 | 76.40±0.13 | 78.71±0.09 | 79.81±0.06 |
| 128 | | | | | 52.59±0.15 | 35.77±0.17 | 72.27±0.09 | 70.97±0.15 | 76.64±0.07 | 78.91±0.04 |
| 196 | | | | | 36.98±0.06 | 8.16±0.09 | 64.28±0.15 | 62.94±0.11 | 74.64±0.13 | 77.99±0.06 |
| 256 | | | | | | 4.13±0.06 | 54.29±0.17 | 53.52±0.08 | 72.59±0.05 | 77.19±0.10 |
| 512 | | | | | | | 4.86±0.10 | 8.11±0.05 | 61.71±0.13 | 73.44±0.05 |
| 784 | | | | | | | 0.79±0.02 | 0.59±0.02 | 45.39±0.11 | 69.27±0.09 |
| 1024 | | | | | | | | 0.41±0.01 | 28.38±0.15 | 65.63±0.10 |
| 2048 | | | | | | | | | 0.65±0.02 | 49.99±0.14 |
| 3136 | | | | | | | | | 0.24±0.01 | 33.89±0.10 |
| 4096 | | | | | | | | | | 21.45±0.09 |
| 8192 | | | | | | | | | | 0.92±0.02 |
| 12544 | | | | | | | | | | 0.29±0.01 |

Figure 15: Performance results of ViT-B-16 model against patch shuffling of the different degrees. Each cell represents the mean and standard deviation of classification accuracy for 5 runs with different random seeding.

**Performance of ViT-B-16 model on ImageNet2012 with "patch blackening" SSDI corruptions**
*(clean accuracy: 81.07%, original image size: 224x224)*

| Num Corrupted Patches | 4 patches of 112x112 | 16 patches of 56x56 | 49 patches of 32x32 | 64 patches of 28x28 | 196 patches of 16x16 | 256 patches of 14x14 | 784 patches of 8x8 | 1024 patches of 7x7 | 3136 patches of 4x4 | 12544 patches of 2x2 |
|---|---|---|---|---|---|---|---|---|---|---|
| 0 | 81.07 | 81.07 | 81.07 | 81.07 | 81.07 | 81.07 | 81.07 | 81.07 | 81.07 | 81.07 |
| 1 | 77.49±0.08 | 80.41±0.05 | 80.78±0.03 | 80.83±0.05 | 80.95±0.03 | 80.93±0.03 | 80.98±0.02 | 80.95±0.02 | 80.98±0.03 | 81.00±0.01 |
| 2 | 69.16±0.09 | 79.55±0.08 | 80.49±0.07 | 80.62±0.06 | 80.84±0.02 | 80.82±0.07 | 80.97±0.02 | 80.86±0.02 | 80.96±0.03 | 80.99±0.03 |
| 4 | 0.10±0.00 | 77.26±0.10 | 79.75±0.08 | 80.26±0.07 | 80.65±0.04 | 80.67±0.06 | 80.88±0.03 | 80.77±0.06 | 80.85±0.05 | 80.96±0.02 |
| 16 | | 0.10±0.00 | 75.19±0.14 | 76.50±0.09 | 79.75±0.11 | 79.44±0.04 | 80.39±0.05 | 80.01±0.06 | 80.41±0.02 | 80.61±0.05 |
| 32 | | | 61.16±0.08 | 67.09±0.06 | 78.60±0.05 | 77.41±0.13 | 79.65±0.02 | 79.16±0.08 | 79.94±0.06 | 80.43±0.06 |
| 49 | | | 0.10±0.00 | 43.36±0.12 | 77.27±0.09 | 74.69±0.11 | 78.89±0.06 | 78.20±0.10 | 79.51±0.06 | 80.24±0.03 |
| 64 | | | | 0.10±0.00 | 75.86±0.07 | 71.54±0.08 | 78.19±0.09 | 77.33±0.10 | 79.13±0.05 | 79.97±0.04 |
| 128 | | | | | 61.75±0.12 | 46.11±0.26 | 74.50±0.11 | 73.14±0.09 | 77.60±0.04 | 79.30±0.05 |
| 196 | | | | | 0.10±0.00 | 9.08±0.06 | 69.02±0.03 | 67.53±0.17 | 75.87±0.05 | 78.61±0.11 |
| 256 | | | | | | 0.10±0.00 | 62.21±0.15 | 61.68±0.07 | 74.24±0.10 | 77.90±0.04 |
| 512 | | | | | | | 16.21±0.10 | 22.80±0.16 | 66.59±0.10 | 75.12±0.05 |
| 784 | | | | | | | 0.10±0.00 | 0.78±0.03 | 56.67±0.09 | 72.40±0.09 |
| 1024 | | | | | | | | 0.10±0.00 | 46.28±0.14 | 70.23±0.09 |
| 2048 | | | | | | | | | 4.89±0.09 | 61.76±0.11 |
| 3136 | | | | | | | | | 0.10±0.00 | 52.98±0.08 |
| 4096 | | | | | | | | | | 45.19±0.07 |
| 8192 | | | | | | | | | | 9.88±0.04 |
| 12544 | | | | | | | | | | 0.10±0.00 |

Figure 16: Performance results of ViT-B-16 model against patch blackening of the different degrees. Each cell represents the mean and standard deviation of classification accuracy for 5 runs with different random seeding.

| Performance of ResNet50 model on ImageNet2012 with "patch shuffling" SSDI corruptions | | | | | | | | | |
|---|---|---|---|---|---|---|---|---|---|
| (clean accuracy: 80.85%, original image size: 224x224) | | | | | | | | | |
| Patch Size | | | | | | | | | |
| 4 patches of 112x112 | 16 patches of 56x56 | 49 patches of 32x32 | 64 patches of 28x28 | 196 patches of 16x16 | 256 patches of 14x14 | 784 patches of 8x8 | 1024 patches of 7x7 | 3136 patches of 4x4 | 12544 patches of 2x2 |

**Number of Corrupted Patches**

| | 4 patches of 112x112 | 16 patches of 56x56 | 49 patches of 32x32 | 64 patches of 28x28 | 196 patches of 16x16 | 256 patches of 14x14 | 784 patches of 8x8 | 1024 patches of 7x7 | 3136 patches of 4x4 | 12544 patches of 2x2 |
|---|---|---|---|---|---|---|---|---|---|---|
| 0 | 80.85 | 80.85 | 80.85 | 80.85 | 80.85 | 80.85 | 80.85 | 80.85 | 80.85 | 80.85 |
| 2 | 77.74±0.04 | 79.27±0.03 | 80.21±0.06 | 80.38±0.02 | 80.67±0.04 | 80.71±0.02 | 80.75±0.04 | 80.75±0.03 | 80.81±0.04 | 80.83±0..04 |
| 4 | 77.42±0.08 | 77.57±0.08 | 79.56±0.08 | 79.89±0.07 | 80.53±0.07 | 80.56±0.05 | 80.70±0.03 | 80.68±0.03 | 80.74±0.02 | 80.77±0.02 |
| 16 | | 71.38±0.06 | 74.45±0.06 | 76.11±0.06 | 79.39±0.08 | 79.72±0.09 | 80.05±0.06 | 80.18±0.07 | 80.29±0.03 | 80.42±0.06 |
| 32 | | | 64.00±0.07 | 68.01±0.16 | 77.32±0.13 | 78.05±0.06 | 79.13±0.111 | 79.20±0.07 | 79.55±0.06 | 79.81±0.08 |
| 49 | | | 57.69±0.09 | 57.80±0.09 | 74.06±0.08 | 75.73±0.10 | 77.96±0.07 | 77.95±0.07 | 78.60±0.05 | 79.01±0.06 |
| 64 | | | | 53.73±0.18 | 69.97±0.09 | 72.93±0.11 | 76.70±0.08 | 76.79±0.10 | 77.64±0.09 | 78.33±0.07 |
| 128 | | | | | 39.43±0.20 | 51.29±0.16 | 69.55±0.12 | 70.16±0.10 | 73.41±0.08 | 75.45±0.06 |
| 196 | | | | | 19.23±0.05 | 19.16±0.10 | 57.73±0.16 | 59.92±0.07 | 67.97±0.15 | 72.33±0.07 |
| 256 | | | | | | 11.14±0.13 | 43.44±0.18 | 48.38±0.17 | 62.67±0.11 | 69.77±0.07 |
| 512 | | | | | | | 2.34±0.04 | 6.00±0.06 | 38.39±0.13 | 58.92±0.14 |
| 784 | | | | | | | 0.67±0.02 | 0.62±0.02 | 18.75±0.10 | 49.04±0.09 |
| 1024 | | | | | | | | 0.44±0.01 | 8.05±0.04 | 42.23±0.04 |
| 2048 | | | | | | | | | 0.41±0.01 | 26.55±0.16 |
| 3136 | | | | | | | | | 0.27±0.01 | 16.39±0.04 |
| 4096 | | | | | | | | | | 8.75±0.11 |
| 8192 | | | | | | | | | | 0.37±0.01 |
| 12544 | | | | | | | | | | 0.22±0.01 |

Figure 17: Performance results of ResNet-50 model against patch shuffling of the different degrees. Each cell represents the mean and standard deviation of classification accuracy for 5 runs with different random seeding.

| Performance of ResNet50 model on ImageNet2012 with "patch blackening" SSDI corruptions | | | | | | | | | |
|---|---|---|---|---|---|---|---|---|---|
| (clean accuracy: 80.85%, original image size: 224x224) | | | | | | | | | |
| Patch Size | | | | | | | | | |

**Number of Corrupted Patches**

| | 4 patches of 112x112 | 16 patches of 56x56 | 49 patches of 32x32 | 64 patches of 28x28 | 196 patches of 16x16 | 256 patches of 14x14 | 784 patches of 8x8 | 1024 patches of 7x7 | 3136 patches of 4x4 | 12544 patches of 2x2 |
|---|---|---|---|---|---|---|---|---|---|---|
| 0 | 80.85 | 80.85 | 80.85 | 80.85 | 80.85 | 80.85 | 80.85 | 80.85 | 80.85 | 80.85 |
| 1 | 77.87±0.09 | 80.33±0.01 | 80.64±0.05 | 80.71±0.03 | 80.77±0.02 | 80.79±0.01 | 80.80±0.02 | 80.81±0.03 | 80.82±0.01 | 80.81±0.01 |
| 2 | 71.65±0.13 | 79.64±0.10 | 80.52±0.04 | 80.57±0.06 | 80.73±0.04 | 80.75±0.04 | 80.76±0.02 | 80.77±0.04 | 80.80±0.02 | 80.81±0.03 |
| 4 | 0.10±0.00 | 78.15±0.07 | 80.18±0.07 | 80.37±0.06 | 80.65±0.02 | 80.65±0.03 | 80.73±0.04 | 80.72±0.03 | 80.74±0.04 | 80.76±0.02 |
| 16 | | 0.10±0.00 | 77.50±0.13 | 78.47±0.11 | 80.13±0.05 | 80.28±0.05 | 80.38±0.05 | 80.35±0.04 | 80.32±0.04 | 80.36±0.03 |
| 32 | | | 67.80±0.21 | 73.84±0.16 | 79.32±0.10 | 79.59±0.05 | 79.88±0.04 | 79.79±0.05 | 79.76±0.04 | 79.71±0.07 |
| 49 | | | 0.10±0.00 | 59.62±0.14 | 78.18±0.01 | 78.81±0.03 | 79.24±0.09 | 79.06±0.04 | 78.93±0.09 | 79.01±0.08 |
| 64 | | | | 0.10±0.00 | 76.87±0.08 | 77.95±0.12 | 78.54±0.15 | 78.41±0.06 | 78.19±0.06 | 78.40±0.07 |
| 128 | | | | | 65.77±0.09 | 71.85±0.04 | 75.17±0.15 | 74.89±0.08 | 75.09±0.10 | 76.23±0.10 |
| 196 | | | | | 0.10±0.00 | 54.61±0.27 | 70.15±0.09 | 69.80±0.14 | 71.58±0.18 | 74.19±0.08 |
| 256 | | | | | | 0.10±0.00 | 63.87±0.11 | 63.85±0.07 | 68.34±0.06 | 72.75±0.06 |
| 512 | | | | | | | 28.53±0.17 | 28.56±0.15 | 54.30±0.12 | 66.56±0.13 |
| 784 | | | | | | | 0.10±0.00 | 4.60±0.09 | 41.46±0.14 | 60.97±0.11 |
| 1024 | | | | | | | | 0.10±0.00 | 30.56±0.16 | 57.61±0.09 |
| 2048 | | | | | | | | | 2.22±0.06 | 53.10±0.12 |
| 3136 | | | | | | | | | 0.10±0.00 | 46.58±0.08 |
| 4096 | | | | | | | | | | 38.46±0.14 |
| 8192 | | | | | | | | | | 5.45±0.11 |
| 12544 | | | | | | | | | | 0.10±0.00 |

Figure 18: Performance results of ResNet-50 model against patch blackening of the different degrees. Each cell represents the mean and standard deviation of classification accuracy for 5 runs with different random seeding.

# D Extra quantitative results

In Table 7 and Table 10 we add more results for the detection task of SSDI corruptions, on CelebA (7 classes) and SUN-RGBD (13 classes), respectively.

**CelebA** The segmentation model is ResNet-18 with FPN. The input images have sizes $256 \times 256$. All results are reported using the first grammar validation method, which relies on the combination of using bi-LSTM and comparing the predicted part semantics to the estimated part semantics within the image itself (i.e. without the memorized average part semantics). From Table 7, we observe that the proposed approach exhibits reliable performance when the size of the corrupted patch exceeds $30 \times 30$: $>80\%$ on patch shuffling SSDI, 92.51% on puzzle solving SSDI with 4 permutations, $>80\%$ on patch blackout SSDI.

**SUN-RGBD** The segmentation model is ResNet-50 with encoder-decoder (RedNet). The input images have sizes $640 \times 480$. All results are reported using the first grammar validation method, which relies on the combination of using bi-LSTM and comparing the predicted part semantics to the estimated part semantics within the image itself (i.e. without the memorized average part semantics). Two segmentation granularities are used, 13-class coarse segmentation merged from 37-classes, and the original 37-class fine segmentation. From Table 10, we observe SSDI detection performance for the two levels of granularity. For patch shuffling SSDI and puzzle solving SSDI, using coarser 13-class semantic masks boosts syntax sequence reasoning and improves grammar validation performance. This is due to the better sequential reasoning capability that the bi-LSTM learns from coarse patch masks. Meanwhile, in the task of patch blackout, using finer 37-class semantic masks has an edge. The blackened patch semantics is easier to discern when the segmentation is finer, leading to a longer syntax sequence and higher variations in the predicted semantic values.

Table 7: SSDI detection performance on CelebA with 256x256 sized images.

| SSDI Corruption: patch shuffling | | | | | |
|---|---|---|---|---|---|
| Test Accuracy (%) | Patch size: 10 | Patch size: 20 | Patch size: 30 | Patch size: 40 | Patch size:50 |
| shuffle 2 patches | 53.86 (75.58) | 65.66 (68.67) | 76.22 (79.72) | 83.89 (86.33) | 88.22 (93.04) |
| shuffle 3 patches | 55.67 (75.16) | 73.16 (78.97) | 85.68 (91.08) | 90.96 (95.22) | 93.05 (97.42) |
| shuffle 4 patches | 57.33 (74.73) | 78.40 (82.29) | 89.79 (94.57) | 93.00 (97.34) | 94.34 (98.07) |
| shuffle 5 patches | 59.13 (70.93) | 82.88 (88.15) | 91.72 (96.06) | 93.93 (97.25) | 94.69 (97.39) |
| SSDI Corruption: puzzle solving | | | | | |
| Test Accuracy (%) | Patch size: 10 | Patch size: 20 | Patch size: 30 | Patch size: 40 | Patch size:50 |
| 1 real, 3 fake | 57.84 | 86.74 | 92.51 | 93.97 | 94.81 |
| 1 real, 119 fake (all perm) | 8.39 | 37.43 | 66.15 | 80.37 | 87.23 |
| SSDI Corruption: patch blackening | | | | | |
| Test Accuracy (%) | Patch size: 10 | Patch size: 20 | Patch size: 30 | Patch size: 40 | Patch size:50 |
| blacken 1 patch | 54.73 (55.36) | 70.91 (67.04) | 81.25 (81.08) | 87.37 (88.04) | 91.25 (95.12) |
| blacken 2 patches | 60.43 (63.78) | 83.83 (85.22) | 91.77 (93.43) | 95.41 (97.41) | 95.78 (97.58) |
| blacken 3 patches | 65.79 (68.94) | 90.60 (93.83) | 96.48 (98.49) | 97.83 (98.57) | 98.50 (99.14) |
| blacken 4 patches | 71.85 (74.34) | 94.56 (97.14) | 98.08 (98.88) | 98.86 (99.36) | 99.05 (99.49) |
| blacken 5 patches | 77.32 (85.28) | 97.22 (98.20) | 98.87 (99.48) | 98.95 (99.29) | 99.13 (99.64) |

Table 8: 13 segmentation classes

**SSDI Corruption: patch shuffling**

| Test Accuracy (%) | Patch size: 160 | Test Accuracy (%) | Patch size: 80 |
|---|---|---|---|
| shuffle 4 patches | 60.57 (72.04) | shuffle 16 patches | 76.57 (73.37) |
| shuffle 8 patches | 69.31 (72.59) | shuffle 32 patches | 86.63 (82.38) |
| shuffle 12 patches | 73.47 (77.50) | shuffle 48 patches | 87.09 (82.97) |

**SSDI Corruption: puzzle solving**

| Test Accuracy (%) | Patch size: 160 | | Patch Size: 80 |
|---|---|---|---|
| 1 real, 3 fake | 72.89 | | 91.17 |
| 1 real, 99 fake | 28.95 | | 69.67 |

**SSDI Corruption: patch blackening**

| Test Accuracy (%) | Patch size: 160 | Test Accuracy (%) | Patch size: 80 |
|---|---|---|---|
| blacken 4 patches | 66.55 (67.21) | blacken 16 patches | 67.43 (65.53) |
| blacken 8 patches | 66.59 (67.17) | blacken 32 patches | 75.68 (70.22) |
| blacken 12 patches | 66.75 (67.49) | blacken 48 patches | 73.66 (74.81) |

Table 9: 37 segmentation classes

**SSDI Corruption: patch shuffling**

| Test Accuracy (%) | Patch size: 160 | Test Accuracy (%) | Patch size: 80 |
|---|---|---|---|
| shuffle 4 patches | 62.32 (74.57) | shuffle 16 patches | 67.74 (74.77) |
| shuffle 8 patches | 71.53 (76.99) | shuffle 32 patches | 78.04 (76.20) |
| shuffle 12 patches | 74.93 (77.43) | shuffle 48 patches | 79.84 (78.69) |

**SSDI Corruption: puzzle solving**

| Test Accuracy (%) | Patch size: 160 | | Patch Size: 80 |
|---|---|---|---|
| 1 real, 3 fake | 76.10 | | 81.23 |
| 1 real, 99 fake | 33.85 | | 55.99 |

**SSDI Corruption: patch blackening**

| Test Accuracy (%) | Patch size: 160 | Test Accuracy (%) | Patch size: 80 |
|---|---|---|---|
| blacken 4 patches | 61.58 (71.37) | blacken 16 patches | 61.41 (71.25) |
| blacken 8 patches | 67.53 (72.20) | blacken 32 patches | 69.07 (71.80) |
| blacken 12 patches | 66.18 (76.00) | blacken 48 patches | 64.59 (72.24) |

Table 10: SSDI detection performance on SUN-RGBD with 640x480 sized images.

# E Extra qualitative results

## E.1 Semantic clustering and segmentation

Figure 19 and Figure 20 show selected part semantics obtained on CelebA face images and SUN-RGBD room scene images, respectively. All samples are from the train set. The displayed part semantics masks are generated by trained ResNet-18 and trained ResNet-50 encoder-decoder, respectively.

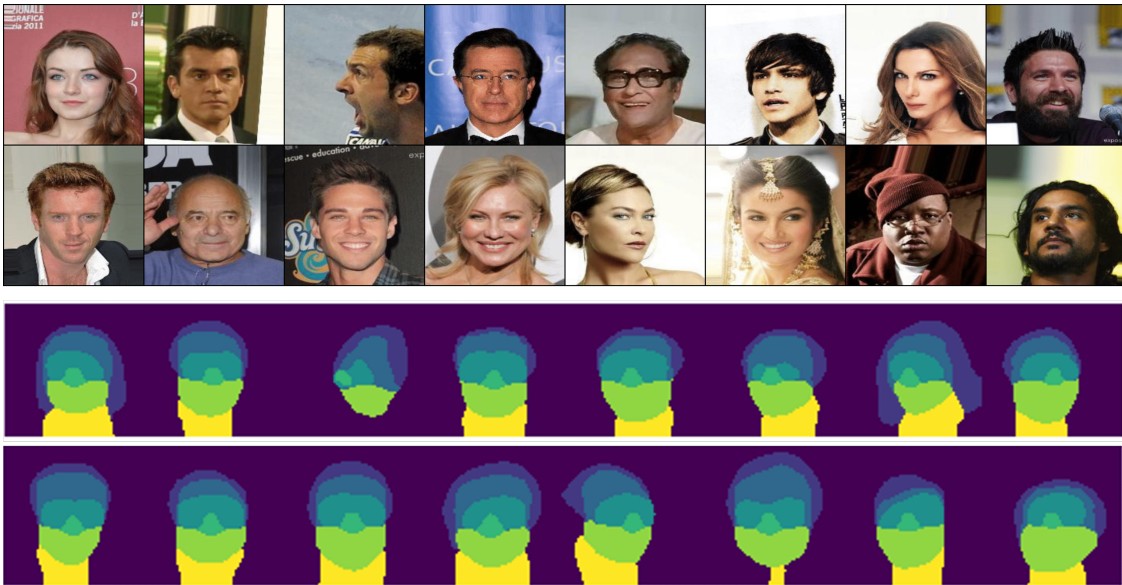

Figure 19: 7-class semantic segmentation on CelebA

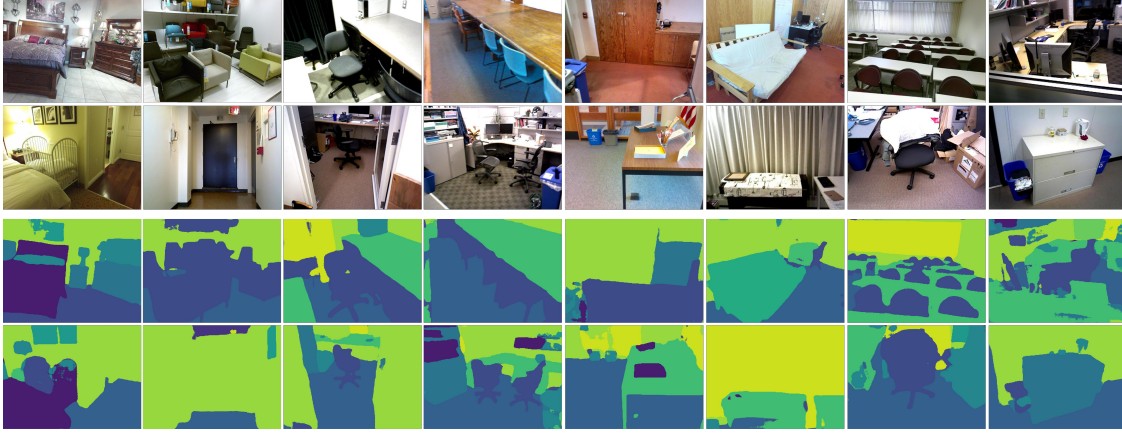

Figure 20: 13-class semantic segmentation on SUN-RGBD

## E.2 Syntactic sequences of part semantics

Figure 21 shows the pixel-level part semantics masks and the distribution of part semantics in each mask, averaged over correct samples in the entire CelebA test set. During training, we optimized MSE loss between bi-LSTM predictions and the actual semantics inside each patch, in order to model the mean of the part semantics distribution in each patch iteration. Hence, here we show the average traversal patterns learned. It can be seen that face-part semantics transitions and the face syntax in the CelebA dataset are captured by the trained bi-LSTM model. For example, in the first 2 patch iterations, the "forehead" semantics is

dominant. In the third and fourth patches, "mouth" semantics is dominant. While for the last patch, "eyes" semantics is dominant. These part semantics along with transitions of those semantics is key to identifying the presence of SSDI.

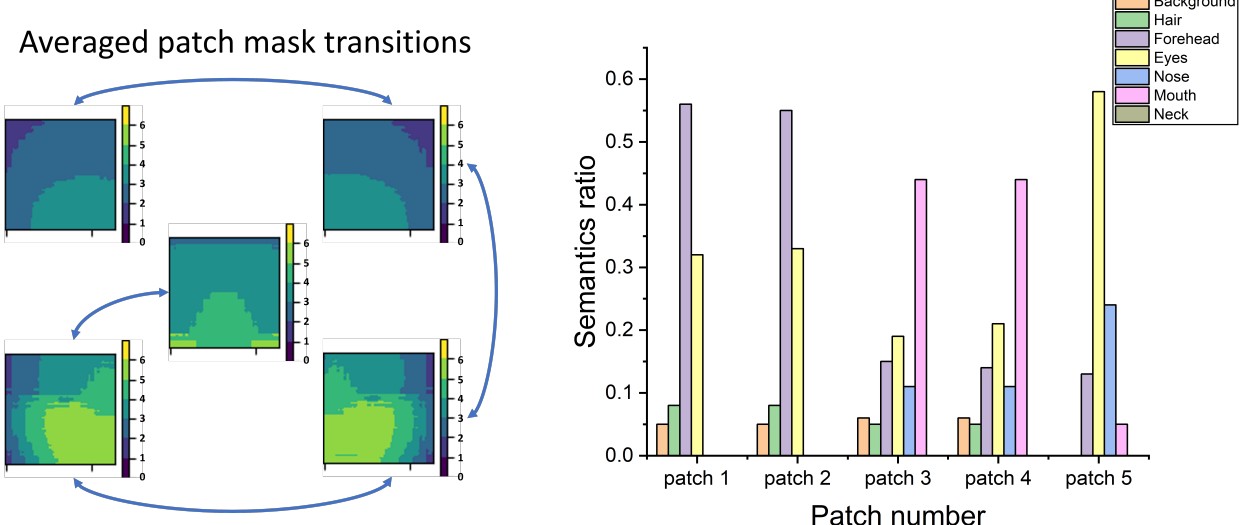

Figure 21: Left: Averaged patch mask transitions over CelebA test set; Right: Averaged patch semantics over CelebA test set

Figure 22 shows the pixel-level part semantics masks and the distribution of part semantics in each mask, averaged over correct samples in the entire SUN-RGBD test set. Here, each of the 13 semantic classes is either a stuff (e.g., ceiling, floor, wall, etc.) or a thing (e.g., books, char, table, etc.). Within the traversal sequence, the first few patches at the top of the image have "wall" as the dominant part semantics, while the last few patches at the bottom have "floor" as the dominant part semantics. Semantic classes belonging to the thing category reside in the middle patches, such as "chair" and "table". This reveals that our proposed framework is capable of learning the syntax of the scene-centric images containing stuff and thing layouts even from a diverse SUN-RGBD dataset with a wide variety of different room scenes.

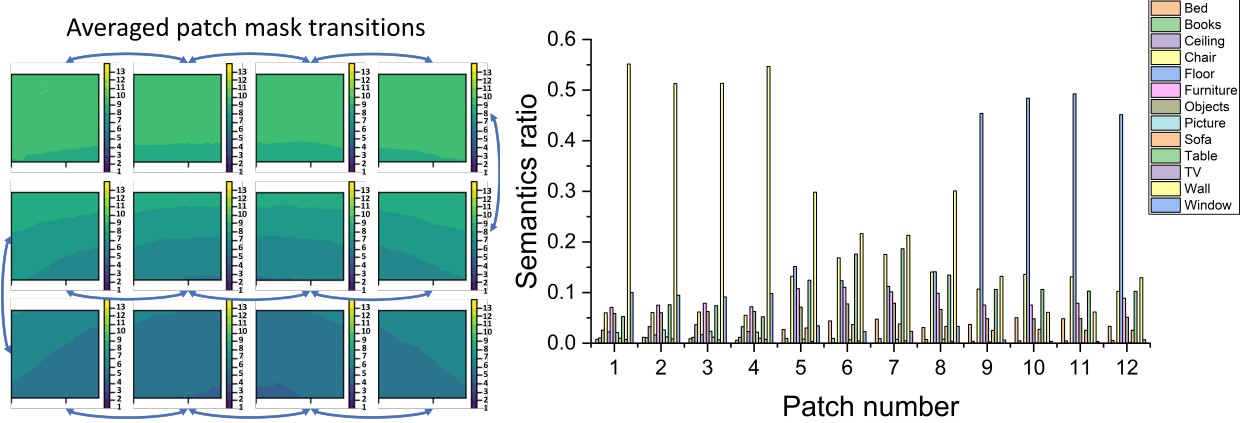

Figure 22: Left: Averaged patch mask transitions over SUN-RGBD test set; Right: Averaged patch semantics over SUN-RGBD test set

