# OpenReview forum: "Semantic-Syntactic Discrepancy in Images (SSDI): Learning Meaning and Order of Features from Natural Images"
_TMLR — Accepted by TMLR_

### Review · Reviewer_ftNh · 2024-12-21

**Summary Of Contributions:**

The paper highlights a new limitation of existing classification models: they tend to overlook changes in the arrangement of object parts, provided that all parts are present, even in an unnatural random order. To figure out this overlooked behavior of the existing classification model, this paper introduces the concept of Semantic and Syntactic Discrepancies in Images (SSDI), framing it as a vulnerability to corruptions that involve removing or shuffling image patches or presenting images in the form of puzzles. To address this vulnerability, the authors propose a novel method to detect SSDI by analyzing the arrangement of image components using a semi-supervised two-stage approach. The efficacy of the proposed method is validated through evaluations of two benchmark datasets.

**Audience:**

Yes

**Broader Impact Concerns:**

To address ethical considerations, it is recommended that CelebA corruptions be visualized using synthetic or generated images instead of real images from the CelebA dataset. This approach mitigates privacy concerns and ensures the ethical use of data, particularly when dealing with sensitive facial imagery.

**Claims And Evidence:**

Yes

**Requested Changes:**

I believe the weaknesses related to the Persuasiveness of SSDI Vulnerability, Evaluation of Existing Classification Models, and Baseline Comparisons are critical for recommending acceptance, as they address the problem's significance and the proposed method's effectiveness. Without addressing these points, it becomes challenging to justify the need to solve this problem and identify effective strategies for tackling it. The other weaknesses, such as scalability, would enhance the work but are not essential for acceptance.

**Strengths And Weaknesses:**

**Strengths**
- The paper is easy to read.
- The finding of SSDI vulnerability is both novel and intriguing.

**Weaknesses**
- *Evaluation of Existing Classification Models*: The paper lacks a comprehensive analysis of the extent to which current classification models suffer from various types of corruption, particularly those linked to SSDI. Quantifying the performance degradation of existing models across different corruption types—such as patch shuffling, blackening, or puzzle-like transformations—would provide a more apparent context for the significance of SSDI vulnerabilities. Additionally, such analysis could help identify the most critical corruption types to address and establish whether the proposed method offers a tangible improvement over current approaches. Without this evaluation, the impact and necessity of the proposed solution remain unclear.
- *Persuasiveness of SSDI Vulnerability*: While SSDI vulnerability is an interesting concept, the argument for its significance is not entirely convincing. Classification models often assume that input images are curated and verified. The frequency and relevance of corrupted data, as simulated by experiments involving patch shuffling or puzzle solving, remain unclear. Furthermore, accurate classification is often crucial in some instances despite corruptions such as patch blackening. These types of corruption appear somewhat artificial and may not represent common real-world scenarios. For instance, it is arguably more critical for models to maintain classification accuracy even under common corruptions rather than addressing artificially generated SSDI cases. Thus, the real-world importance of addressing SSDI vulnerabilities is insufficiently demonstrated.
- *Baseline Comparisons*: Although SSDI vulnerability is not yet well-established, including baseline comparisons would strengthen the validity of the proposed method. For instance, the authors could employ VLM/LLM to detect corrupted images by identifying vulnerabilities or transformations rather than solely classifying the object. Measuring VLM/LLM performance in explaining and detecting transformations, alongside evaluating their failure rates and error patterns, would provide a stronger foundation for comparison.
- *Scalability*: The proposed method's scalability to more complex datasets raises concerns. As datasets increase in complexity—such as moving from CelebA to SUN-RGBD or ImageNet—the distinction between corrupted and original images diminishes. CelebA, with its relatively simple structure (e.g., centered faces), is less challenging than datasets involving intricate and diverse scenes. This calls into question the feasibility of applying the current approach to more complex datasets without significant adaptations.

---

> ### Author Response · Authors · 2025-02-21
> **Response (1) to the initial review by the Reviewer ftNh**
>
> We would like to thank the Reviewer ftNh for their time and valuable feedback. Please find below the list of clarifications and changes we incorporated to the updated manuscript to improve its quality based on your suggestions.
>
> ### **Evaluation of existing classification models**
> Thank you for your suggestion to evaluate existing classification models. The performance results of ViT-B-16 against different types of SSDI patch manipulations are presented in **Section 2.2.1 and Section 2.3** of the updated manuscript. Similar results for ResNet50 model are included in the **Appendix C**. We believe that this set of results on classification models not only highlighted its presence, but also helps to improve the definition of SSDI as a problem. In particular, that there is a gray-area in the performance results, where, despite significant changes to the input, the response of classification models experiences an insignificant impact with respect to the base accuracy.
>
> ### **Baseline comparisons**
> Thank you for the suggestion to incorporate a study of a vision-language model (VLM). We updated the manuscript to include an evaluation of Meta's LLaMA model. Specifically, we used LLaMA version 3.2 with 11 billion parameters available on the HuggingFace hub. We generated and processed various corruption types on 3k samples subset randomly picked from ImageNet2012 dataset. Three various prompts were used, each corresponding to a different level of guidance in revealing the possibility of abnormalities being present in the given images. This would ensure that the model was fairly evaluated in terms of their capability of identifying vulnerabilities. **Section 2.2.2 and Section 2.2.3 in the updated manuscript** provide details on the experiments as well as the corresponding findings. The main findings suggest that, while VLMs are highly capable of detecting patch corruptions when the prompt directly asks about their presence, SSDI corruptions still pose a vulnerability to VLMs when their presence is  not addressed explicitly via careful prompt. This is further supported by their inability to determine the original natural image in the case of puzzle solving SSDI.

---

> ### Author Response · Authors · 2025-02-21
> **Response (2) to the initial review by the Reviewer ftNh**
>
> ### **Persuasiveness of SSDI vulnerability**
> We appreciate the reviewer’s insightful comments and acknowledge the concerns regarding (1) the assumption that classification models operate on curated images, (2) the unclear frequency of SSDI-type corruptions in real-world settings, (3) the continued importance of classification accuracy despite corruptions, and (4) the perceived artificial nature of the SSDI corruptions used in this study. Below, we clarify these points and further justify the real-world relevance of SSDI vulnerability.
>
> #### **Regarding (1): The Assumption of Curated Input Data**
> It is true that classification models are typically trained on curated datasets. However, this very assumption enforces a reliance on the mere **presence of features**, without an explicit understanding of their **natural arrangement**. As a result, models trained solely on natural images lack an inherent mechanism to distinguish between naturally occurring and artificially manipulated images. This leads to high-confidence misclassifications when confronted with unusual but plausible compositions.
>
> During deployment, one could argue that system-level safeguards (e.g., human verification) may help filter unnatural images before they reach classification models. However, such safeguards only attempt to mitigate failures **after the fact**, rather than addressing the fundamental issue—**the model's inability to detect unnatural image compositions autonomously**. In high-stakes applications (e.g., facial recognition, medical imaging, autonomous driving), it is critical for models to possess an internal mechanism to assess image validity rather than relying on external interventions.
>
> #### **Regarding (2) and (4): Real-World Occurrence of SSDI Corruptions**
> While SSDI corruptions in our study were artificially generated, the underlying phenomenon—misalignment, occlusion, or loss of object structure—is not artificial. Our experiments demonstrate that even small disruptions in image structure lead to high-confidence classifications, suggesting that SSDI vulnerability is an inherent weakness of current models.
>
> Beyond our synthetic examples, SSDI-type corruptions occur naturally in various real-world settings:
>
> - **Facial Recognition & Security Systems:** Occlusions due to masks, accessories, or partial obstructions can lead to misidentifications, as models often ignore feature placement while relying solely on feature presence.
> - **Medical Imaging:** Structural anomalies, missing regions in scans (due to artifacts or occlusions), or synthesized images (e.g., AI-generated radiology scans) may be classified incorrectly, posing risks in clinical decision-making.
> - **Autonomous Driving:** Partial occlusions of objects (e.g., pedestrians behind poles, damaged traffic signs) may not prevent models from classifying them, but the loss of structural coherence could lead to dangerous misinterpretations.
>
> Figures 7 and 14 in the manuscript highlight how shuffled facial features are still classified confidently as valid faces, reinforcing the concern that models lack a mechanism to assess the structural validity of images.
>
> #### **Regarding (3): Impact on Classification Accuracy**
> Our proposed approach does not diminish classification performance but instead enhances it by incorporating structural awareness. The goal is not to prevent models from making predictions but rather to enable them to differentiate between valid and structurally inconsistent images.
>
> This issue is especially evident in **vision-language models (VLMs)**: as our experiments show, when given a generic descriptive prompt, these models failed to **explicitly recognize abnormalities** in more than half of the cases. This suggests that models are not inherently trained to detect structural inconsistencies, even when tasked with analyzing images beyond classification.
>
> By incorporating structural validation mechanisms, models could improve their robustness, ensuring that predictions are not only accurate but also semantically and syntactically valid.
>
> We have incorporated the discussed points into **Section 2.3 of the updated manuscript**.

---

### Review · Reviewer_f9ix · 2024-12-24

**Summary Of Contributions:**

The work identifies a vulnerability in common image classification models. Namely, such models will mistakenly classify an object as belonging to a particular category, even if the respective object has been broken up and reshuffled in the image to form an unnatural arrangement. The reason such models still confidently classify the reorganized image as belonging to the original category, is that they are trained to make predictions based on constituent features, irrespective of the order in which such features are present (essentially, the models have no understanding of “image syntax” only “image semantics”.) The authors term this vulnerability Semantic Syntactic Discrepancy in Images (SSDI).

To resolve the SSDI issue in available vision models, the authors leverage an architecture meant to explicitly encourage learning of expected “image syntax” (or expected order of image parts) such that out of order images can be more readily identified. Image grammar is learned in a two stage method;  the first stage is a DNN feature extractor that learns “image semantics” and the second stage is a bi-directional LSTM that learns “image syntax” (the order in which constituent image parts appear heuristically).

The authors propose to evaluate a model for its ability to handle SSDI appropriately, by creating a dataset of both naturally occurring and unnatural images (e.g. shuffled patches). In order to enable the model to infer whether an image is syntactically in distribution or out of grammatically reasonable distribution, the authors establish 3 methods of quantifying how likely an image is to belong to a natural set of images. The methods include 1) computing the average difference between predicted part semantics from the syntax aware bi-directional LSTM vs. the part semantics predicted by the first stage DNN, 2) computing the average difference between predicted part semantics and average per pixel semantics from the training dataset (like taking the difference between prediction and average expected value) and 3) the same as 2 but using mIoU (mean intersection over union). For a set of corrupted and uncorrupted images, the error values are determined, and an error-value threshold is set with which to classify images as corrupted or uncorrupted during inference.

The authors create 3 types of corruptions, and apply their method to two datasets CelebA and SUN-RGBD. They evaluate the accuracy and recall values, for identifying corrupted vs uncorrupted images in the test datasets, and benchmark against SemanticGAN. The authors find their method able to recall SSDI for 70-90% of corrupted images on these two datasets.

**Audience:**

Yes

**Broader Impact Concerns:**

No concerns, helpful statement as to the potential impact of not correcting SSDI (which the paper highlights as a currently unresolved issue).

**Claims And Evidence:**

Yes

**Requested Changes:**

Please benchmark against Gemini for detecting SSDI. This might require a bit of creativity (e.g. determining what prompt to use for detecting SSDI). As long as the method is well described, that should be sufficient.

Discuss the limitations of the patching method, and how a more robust patching system could be implemented (which would be more robust to scales, types of images, where parts are located, etc.)

Discuss why the proposed method may not be outperforming the two proposed baselines.

Minor

Add the line about how the threshold is calculated

**Strengths And Weaknesses:**

Strengths

Compelling problem identification and exposition of problem (SSDI vulnerability of vision models)

Proposal of a semi-supervised solution, which proposes to leverage bi-directional LSTM architecture that enables additionally grasp expected syntax, in order to identify if images are syntactically out of distribution come inference time

Evaluation on two diverse real world datasets

Thorough evaluation of classification methods (3 variations of dataset corruptions, 3 types of error estimation, and two evaluation criteria of accuracy and recall).

Weaknesses

The 5 patch based method for training the bi-directional LSTM does not seem robust. Use of 5 patches in a very  particular region is quite a fragile methodological proposal. This choice may be quite sensitive to the size of relevant image features, and may need to be tuned for object types of various scales and location placement. A more robust proposal, perhaps with e.g. a greater number / coverage of randomly selected patches, might be more compelling and less tailored to the demonstrative dataset.

Lack of benchmarking results against the highlighted model at beginning of problem statement (i.e. Gemini)

The proposed method does not consistently outperform the baselines (SemanticGAN and Dformer)

Minor

The method for determining threshold value is not explicitly noted (assume it is done using ROC curve, but could not find the sentence in the paper.) If it is not already present somewhere, please include an additional one line note.

---

> ### Author Response · Authors · 2025-02-21
> **Response to the initial review by the Reviewer f9ix**
>
> We would like to thank the Reviewer f9ix for their time and valuable feedback. Please find below the list of clarifications and changes we incorporated to the updated manuscript to improve its quality based on your suggestions.
>
> ### **Benchmarking against vision-language model (VLM)**
> Thank you for the suggestion to incorporate benchmarking results against Gemini model.
> In order to be able to run the model locally on the subset of data, we addressed this by benchmarking Meta's LLaMA vision-language model available on the HuggingFace hub.
>
> Specifically, we used LLaMA version 3.2 with 11 billion parameters to evaluate various corruption types on 3k samples subset randomly picked from ImageNet2012 dataset. Three various prompts were used, each corresponding to a different level of guidance in revealing the possibility of abnormalities being present in the given images. **Section 2.2.2 and Section 2.2.3 in the updated manuscript** provide details on the experiments as well as the corresponding findings. The main findings suggest that, while VLMs are highly capable of detecting SSDI corruptions when the prompt directly asks about their presence, SSDI corruptions still pose a vulnerability to VLMs when their presence is not addressed explicitly via careful prompt. This is further supported by their inability to determine the original natural image in the case of puzzle solving SSDI.
>
> ### **Limitations and more robust patching system**
> Indeed, we acknowledge that the methodology relying on the extraction of fixed patches has its limitations. In fact, the original design of the methodology was aimed to incorporate the mechanisms to predict "where to look" along with the syntax learning. This can be implemented using the methodologies similar to the works of FALcon or GFNet, as was mentioned in the description of **Stage 2 of Section 4** as well as in **the limitations section**. However, due to complexity of the design, we opted out to focus on the fixed patching approach in this exploratory work. We updated the limitation section in the manuscript to reflect on this limitation and the provide more details on how active vision can be used to address it.
>
> ### **Performance comparisons with baselines**
> We appreciate the reviewer’s observation regarding our method’s performance in comparison to SemanticGAN and Dformer. While our approach does not consistently outperform these baselines in all cases, we believe this is due to differences in how segmentation masks interact with different image types and corruption methods.
>
> Notably, when SemanticGAN is used as the part semantics model, our method consistently achieves higher detection performance, regardless of the grammar evaluation method. In contrast, on the SUN-RGBD dataset, our approach and the Dformer-based method show comparable performance.
>
> Although we cannot pinpoint a single definitive reason for this behavior, we suspect it may be related to the nature of the datasets:
>
> - CelebA is object-centric, where a more structured segmentation mask is likely to help in learning the natural arrangement of features.
> - SUN-RGBD is scene-centric, meaning segmentation granularity may play a more significant role in capturing relationships between image patches.
>
> This suggests that the effectiveness of part semantics models is dataset-dependent, and the granularity of segmentation masks may influence how well models learn structural relationships. We have reflected on this observation in the **"Effect of Segmentation Granularity" section of Section 5**.
>
> ### **Determining threshold value**
> Thank you for bringing this to our attention. While we mentioned the threshold in multiple places, we had not provided sufficient details on its selection. We have now addressed this by adding the necessary explanation to **the last paragraph of Section 4**.

---

### Review · Reviewer_np8H · 2025-01-30

**Summary Of Contributions:**

Deep learning models often struggle to recognize unnatural objects that are easily detected by human perception. So, this work highlights the vulnerability of classification models to semantic and syntactic discrepancy in images (SSDI), where object parts are shuffled or removed. To address this, this paper proposes a semi-supervised framework that first (1) extracts patch/pixel-level representations based on deep clustering method, and then (2) summarizes them based on spatial relationships using bi-directional LSTM for detecting unnatural corruptions. In experiments, this paper demonstrates that the proposed framework achieves high SSDI detection performance on CelebA and SUN-RGBD datasets.

**Audience:**

Yes

**Claims And Evidence:**

No

**Requested Changes:**

1. The aforementioned weaknesses.
1. Minor clarification
    - In Eq (2), what is $z_{ipk}$?
    - What is the difference between $z^{(1)}$ and $z^{(2)}$?
    - How to "features of different images should map to the different clusters" using Eq (4)? The objective seems to assign $z_{ip}^{(1)} $  to the cluster $y_{ip}^{(2)}$. How does this assign "features of different images" to the "different clusters"?

**Strengths And Weaknesses:**

### Strengths
1. This paper is generally well-written and easy-to-follow.
1. The proposed framework seems to be extensible, e.g., other backbones.
---
### Weaknesses
1. **SSDI is ill-defined and not well-motivated.** I think SSDI is designed for unnaturalness detection, but "unnaturalness" focused in this paper is unclear. This paper also does not describe a specific application where SSDI can be utilized. I believe that the definition and motivation of SSDI are too vague. In addition, I am wondering what is the advantage of the concept of SSDI over existing unnaturalness detection concepts, including out-of-distribution/anomaly detection and adversarial defense.
1. **The scope of "unnaturalness" covered in this work is very limited.** This paper has mainly focused on "patch shuffling" for creating unnatural images and evaluating models. I think the approach is not enough to detect various plausible unnatural cases. Additionally, the proposed framework seems to able to detect only patch-shuffled or -masked images, so I am not convinced that the framework is widely applicable.
1. **Experiments are limited to SSDI detection evaluation on two datasets.** I think the importance of SSDI detection in real-world applications/scenarios should be demonstrated in the experiment section, but this paper only focuses on evaluating SSDI detection performance on few datasets. I am not convinced that the SSDI detection performance is a meaningful metric for detecting unnaturalness in various real-world applications such as anomaly detection, out-of-distribution detection, etc. In addition, the generalization ability (for unseen datasets, unseen corruptions, etc) should be tested.
1. **Existing pretrained models have not been utilized enough.** In recent years, there are many pre-trained models, especially for dense prediction tasks like object detection or segmentation, for example, Grounding DINO (Liu et al., 2023) and Segment Anything Model (Kirillov et al., 2023). I think, in this problem, it is very crucial to utilize various pretrained models because it is already shown that they can be generalized to various image domains and datasets.

---

> ### Author Response · Authors · 2025-02-21
> **Response (1) to the initial review by the Reviewer np8H**
>
> We would like to thank the Reviewer np8H for their time and valuable feedback. Please find below the list of clarifications and changes we incorporated to the updated manuscript to improve its quality based on your suggestions.
>
> ### **Definition and motivation of SSDI**
> We appreciate the reviewer’s feedback and have substantially revised **Section 2** of the manuscript to clarify the definition of Semantic-Syntactic Discrepancy in Images (SSDI). Specifically, we clarify the definition of SSDI as follows:
>
> > **"SSDI occurs in images that have _visually identifiable semantic features_ but appear unnatural to humans due to the _incorrect arrangement_ or _missing_ of some of those features."**
>
> This definition distinguishes SSDI from other "unnaturalness" detection concepts, such as Out-of-Distribution (OoD) detection and adversarial robustness, based on their different underlying assumptions and objectives.
>
> #### **SSDI vs. Adversarial Robustness:**
> SSDI and adversarial robustness differ **fundamentally in their purpose and mechanism**:
>
> - Adversarial attacks introduce **subtle pixel perturbations** that degrade model decision-making **while keeping images visually indistinguishable from the original**. Their goal is to bypass any potential human verification step before tricking the model.
> - SSDI manipulations, in contrast, **retain object features but disrupt their natural arrangement**, leading models to confidently misclassify syntactically (visually) corrupted images. This difference makes SSDI particularly relevant in fully automated systems, where models scrape, filter, or process images autonomously (e.g., from the internet) without human oversight.
>
> Thus, while adversarial robustness focuses on deceptive, near-identical perturbations, SSDI targets structural inconsistencies that cause confident model decisions in fully autonomous pipelines for visually erroneous samples.
>
> #### **SSDI vs. out-of-distribution samples:**
> SSDI and OoD detection differ in how they define **unseen distributions**:
>
> - OoD detection typically assumes that images from the test distribution remain **structurally intact**, even if they originate from a different semantic domain (e.g., training on "photos of dogs" vs. testing on "hand-drawn sketches of dogs" or shifting from "dog images" to "cat images").
> - SSDI, however, focuses on images that **share the same semantic features as the training data but in an unnatural syntactic arrangement**—i.e., configurations that are possibly rarely encountered in reality by either humans or AI models.
>
> Thus, SSDI is not simply a subset of OoD detection but rather a complementary problem:
>
> - OoD detection addresses shifts in semantic distributions (different object categories or styles).
> - SSDI addresses shifts in syntactic distributions (unnatural object compositions within the same categories).
>
> By making this distinction clearer in our revised manuscript, we aim to emphasize SSDI's unique role in robustness research as a challenge distinct from, yet complementary to, both adversarial robustness and OoD detection.

---

> ### Author Response · Authors · 2025-02-21
> **Response (2) to the initial review by the Reviewer np8H**
>
> ### **Applicability and coverage of SSDI Corruptions**
> While SSDI corruptions in our study were artificially generated, the underlying phenomenon—misalignment, occlusion, or loss of object structure—is not artificial. Experiments in Section 2 of the updated manuscript demonstrate that even small disruptions in image structure lead to high-confidence classifications, suggesting that SSDI vulnerability is an inherent weakness of current models.
>
> Beyond our synthetic examples, SSDI-type corruptions may occur naturally in various real-world settings:
>
> - **Facial Recognition & Security Systems:** Occlusions due to masks, accessories, or partial obstructions can lead to misidentifications, as models often ignore feature placement while relying solely on feature presence.
> - **Medical Imaging:** Structural anomalies, missing regions in scans (due to artifacts or occlusions), or synthesized images (e.g., AI-generated radiology scans) may be classified incorrectly, posing risks in clinical decision-making.
> - **Autonomous Driving:** Partial occlusions of objects (e.g., pedestrians behind poles, damaged traffic signs) may not prevent models from classifying them, but the loss of structural coherence could lead to dangerous misinterpretations.
>
> Figures 7 and 14 in the manuscript highlight how shuffled facial features are still classified confidently as valid faces, reinforcing the concern that models lack a mechanism to assess the structural validity of images.
>
> ### **Assessment of existing pretrained models**
> Based on the reviewers' suggestions, we have expanded **Section 2** to include the evaluation of the variety of existing models (both image-only and vision-and-language) against the range of patch corruptions. Please refer to **Sections 2.2** for the corresponding results. We believe that this set of results on classification models not only highlighted its presence, but also helps to improve the definition of SSDI as a problem. In particular, that there is a gray-area in the performance results, where, despite significant changes to the input, the response of classification models experiences an insignificant impact with respect to the base accuracy.
>
> Additionally, we also trained our model on SUN-RGBD dataset with Grounding DINO and Segment Anything Model (SAM) as part semantics models. The results are shown in the table below. As it can be seen, DINO and SAM can be used to improve the SSDI detection performance when the size of corrupted patches is large. This is due to the observation made for Dformer-S and SemanticGAN baseline models in **"Effect of Segmentation Granularity" paragraph of Section 5**. Specifically, the proposed method can directly benefit from improved segmentation masks, as their accuracy improves the quality of part semantics and hence the learning of their correlation between neighboring patches.
>
> | Dataset            | Part Semantics Model                | Grammar Validation Method | Shuffle 4 160x160 | Shuffle 16 80x80 | Black 4 160x160 |
> | ------------------ | ----------------------------------- | ------------------------- | ----------------- | ---------------- | --------------- |
> | SUN-RGBD (13-cls.) | ResNet50+EncoderDecoder (ours)      | Bi-LSTM + next semantics  | 60.57 (72.04)     | 76.57 (73.37)    | 66.55 (67.21)   |
> | SUN-RGBD (13-cls.) | Dformer-S (Yin et al., 2023) (2023) | Bi-LSTM + next semantics  | 62.16 (73.47)     | 74.23 (72.80)    | 68.86 (71.23)   |
> | SUN-RGBD (13-cls.) | GroundingDINO + GroundedSAM2 (2024) | Bi-LSTM + next semantics  | 92.01 (86.27)     | 74.60 (61.80)    | 93.30 (88.43)   |
>
> _Edit: copy-pasted a wrong data for the second row of the table._

---

> ### Author Response · Authors · 2025-02-24
> **Response (3) to the initial review by the Reviewer np8H**
>
> ### **Minor clarifications**
> We apologize for the delayed response regarding the mathematical clarifications. Equations (2)-(5) describe the process of learning object parts (part semantics) using the PiCIE deep clustering technique (Cho et al., 2021). Because this technique was not originally proposed by us, we provided only a brief description that omitted some details and may have led to confusion. We have now expanded the explanation and the associated mathematics in **Stage 1 of Section 4**.
> Additionally, please note the following clarifications:
> 1. **Clarification on $z_{ipk}$:**
>     The symbol $z_{ipk}$​ represents pixel-level features extracted from the DNN feature extractor, where the indices denote the image $i$, the pixel $p$, and the cluster $k$. Here, $k$ corresponds to the cluster centroid that is closest (in terms of distance) to the pixel-level feature.
> 2. **Clarification on $z^{(1)}$ and $z^{(2)}$:**
>     $z^{(1)}$ and $z^{(2)}$ denote the pixel-level features obtained from two distinct views of the same image. Specifically, as illustrated in Figure 4, the method applies two different sets of transformations to each image, resulting in two sets of features, $z^{(1)}\_{i}$ and $z^{(2)}\_{i}$​. The revised manuscript now includes detailed equations and explanations regarding the generation of these features.
> 3. **Correction of Equation 4 (now Equation 6):**
>     We acknowledge an error in the original description of Equation 4. In the updated manuscript, we clarify that:
>     - The goal of $L_{cross}$​ is _“to train the model so that pixel-level features from two streams are assigned to the same cluster centroid even when different sets of transformations are applied.”_
>     - The goal of $L_{within}$​ is _“to train the model to extract pixel-level features that are consistently assigned to the same cluster centroid when a specific set of transformations is applied.”_
>
> Please refer to the updated **Stage 1 of Section 4** for further details.

---

### Author Response · Authors · 2025-04-28
**Thank you to Reviewers and Action Editor**

Dear Action Editor and Reviewers,

We would like to sincerely thank you for your thoughtful feedback, constructive suggestions, and time throughout the review process!
The insights and recommendations provided were instrumental in helping us improve the quality and clarity of the manuscript.
We are grateful for the opportunity to have our paper accepted for publication at Transactions on Machine Learning Research.

Thank you once again for helping to strengthen our work.

Best regards,

The authors of TMLR submission #3543

---

### Decision · Action_Editor_sAr2 · 2025-03-29

**Recommendation:** Accept as is

**Comment:**

The paper highlights a vulnerability in image classifiers, termed Semantic and Syntactic Discrepancy in Images (SSDI), where models fail to detect unnatural arrangements of object parts, such as patch shuffling or masking. To address this, the authors propose a two-stage framework that captures both semantic content and syntactic structure to detect SSDI corruptions.

All reviewers found the problem formulation interesting and the proposed method clear and well-evaluated on diverse datasets. The authors also successfully addressed the reviewers’ concerns in their revision, leading to unanimous agreement on acceptance. Therefore, the AE recommends acceptance.

**Audience:**

Yes.

**Claims And Evidence:**

Yes.